# Structure of the native myosin filament in the relaxed cardiac sarcomere

Davide Tamborrini[1], Zhexin Wang[1,3], Thorsten Wagner[1], Sebastian Tacke[1], Markus Stabrin[1], Michael Grange[1,4], Ay Lin Kho[2], Martin Rees[2], Pauline Bennett[2], Mathias Gautel[2] & Stefan Raunser[1✉]

The thick filament is a key component of sarcomeres, the basic units of striated muscle[1]. Alterations in thick filament proteins are associated with familial hypertrophic cardiomyopathy and other heart and muscle diseases[2]. Despite the central importance of the thick filament, its molecular organization remains unclear. Here we present the molecular architecture of native cardiac sarcomeres in the relaxed state, determined by cryo-electron tomography. Our reconstruction of the thick filament reveals the three-dimensional organization of myosin, titin and myosin-binding protein C (MyBP-C). The arrangement of myosin molecules is dependent on their position along the filament, suggesting specialized capacities in terms of strain susceptibility and force generation. Three pairs of titin-α and titin-β chains run axially along the filament, intertwining with myosin tails and probably orchestrating the length-dependent activation of the sarcomere. Notably, whereas the three titin-α chains run along the entire length of the thick filament, titin-β chains do not. The structure also demonstrates that MyBP-C bridges thin and thick filaments, with its carboxy-terminal region binding to the myosin tails and directly stabilizing the OFF state of the myosin heads in an unforeseen manner. These results provide a foundation for future research investigating muscle disorders involving sarcomeric components.

Muscle contraction requires a shortening of the sarcomere, the basic contractile unit of muscle, caused by the sliding of interdigitating thick and thin filaments. Thick filaments are bipolar structures containing myosin II, MyBP-C and titin, as well as several other proteins near the bare zone, which marks the symmetry axis of the thick filament and the centre of the sarcomere[3]. By convention, this region comprises the P zone (P stands for proximal), where the so-called crowns of myosin P1, P2 and P3 are located, and the M band, a bare zone devoid of myosin heads where myosin tails belonging to apposing half-sarcomeres intertwine[4]. Myosin is a 530-kDa complex of a myosin II dimer organized into two distinct regions: the heads and the tails. Each head comprises a motor domain associated with an essential light chain and a regulatory light chain. The two tails form a coiled-coil and insert into the thick filament backbone. Myosin heads are arranged in a quasihelical array with a three-fold rotational symmetry in the relaxed state of muscle[5–7]. The pairs of heads of three myosin molecules protrude from the backbone at regular intervals to form a 'crown' of heads, with successive crowns axially separated by about 143 Å. The interaction of the myosin motor domains with the thin filament is responsible for force generation and muscle contraction.

MyBP-C localizes in the central region of the A band, the C zone, which also contains all of the myosin motors that are responsible for providing the peak force during contraction[8]. The cardiac isoform of MyBP-C (cMyBP-C) is a 140-kDa protein with three fibronectin type 3 (FN3)-like domains and eight immunoglobulin-like domains that can

form links between the thick and thin filaments[9]. Biochemical investigations support a model in which the amino-terminal region can bind to either myosin heads or actin in a phosphorylation-dependent manner[10], whereas the C-terminal region docks to the thick filament via interactions with titin and myosin tails[11,12]. MyBP-C links can act as mechanical sensors for muscle tension and have both activating and inhibitory effects on the myosin motors, fine-tuning the strength and the kinetics of the muscle contraction[13,14].

Titin is a protein of up to 4.25 MDa consisting of up to 169 immunoglobulin-like and 132 FN3-like domains that form a chain from the M band to the Z disc, where the actin filaments of opposite polarity are crosslinked by α-actinin, marking the lateral borders of the sarcomeres[15,16]. Titin acts as a molecular spring that prevents the overstretching of the sarcomere, recoils the sarcomere after stretch release[17] and is proposed to act as a molecular ruler for assembly of myosin in the A band (the region of the sarcomere where thin and thick filaments overlap)[18,19]. The exact stoichiometry of titin within the A band has remained a mystery, with estimates ranging from 2 to 10 molecules per thick filament[20]. With numerous post-translational modification sites scattered along its length and the predicted interactions with myosin and MyBP-C, titin is also proposed to mediate regulatory signals from the cell to the myosin motors[21].

Models exist for the general arrangement of myosin heads within the cardiac thick filament[22], and using components reconstituted in vitro,

[1]Department of Structural Biochemistry, Max Planck Institute of Molecular Physiology, Dortmund, Germany. [2]Randall Centre for Cell and Molecular Biophysics, School of Basic and Medical Biosciences, Kings College London BHF Centre of Research Excellence, London, UK. [3]Present address: Structural Studies Division, MRC Laboratory of Molecular Biology, Cambridge, UK. [4]Present address: Structural Biology, The Rosalind Franklin Institute, Didcot, UK. ✉e-mail: stefan.raunser@mpi-dortmund.mpg.de

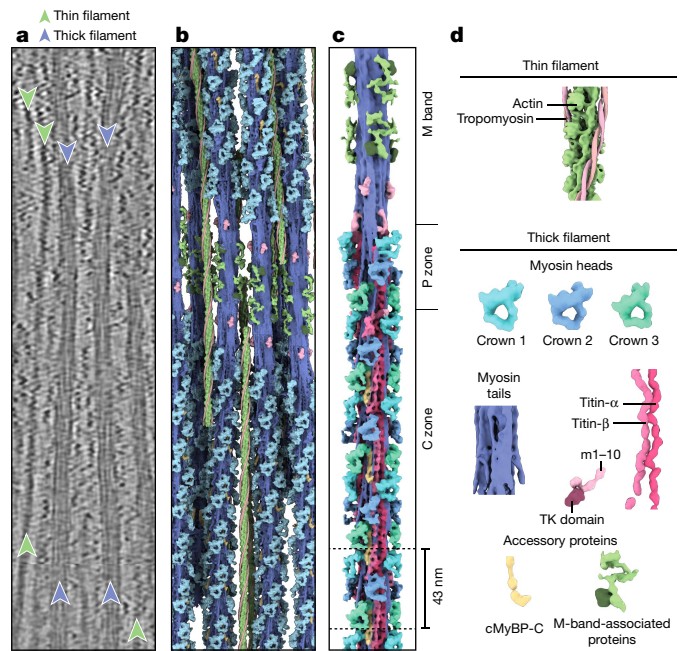

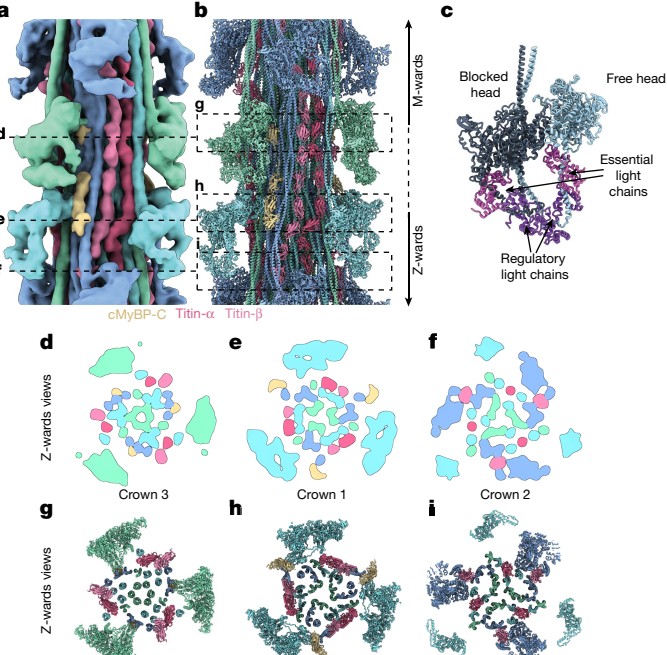

**Fig. 1 | Thick filament structure in the relaxed cardiac sarcomere.**
**a**, Tomographic slice of a cardiac sarcomere, centred on the M band, depicting thick and thin filaments. Scale bar, 50 nm. **b**, Reconstructed thick and thin filaments mapped into a tomogram. Thin filaments obstructing the view on the thick filament were removed for clarity. Scale bar, 50 nm **c**, Structure of the thick filament from the M band to the C zone. For clarity, only the first four cMyBP-C stripes are shown. **d**, Illustration of the various sarcomere components and their colour code, which is maintained throughout the manuscript, unless otherwise indicated. For abbreviations and colour codes for the structural elements of the thick filament, see Extended Data Fig. 2k.

**Fig. 2 | Structural model of the C zone. a,b**, The 3D reconstruction (**a**) and atomic model (**b**) of the C zone, from cMyBP-C stripe no. 4 to stripe no. 9. The volume is coloured according to its atomic model (**b**). **c**, Model of the myosin double heads in the conformational OFF state (Protein Data Bank: 5TBY)[27], the IHM, with the blocked head and the free head binding to each other. **d–i**, Z-ward views of cross-sections of the map (**d–f**) and model (**g–i**) at the level of crown 3 (**d,g**), crown 1 (**e,h**) and crown 2 (**f,i**), providing a more detailed view of the arrangement within the core of the thick filament.

remarkable insights have been gleaned into the regulation of their kinetics[23]. However, how these observations map onto the structure of a fully complemented, natively organized sarcomere and the molecular interactions that occur within the thick filament in its native environment are still unknown. In particular, knowledge of key molecular players implicated in disease—myosin, titin and MyBP-C—is completely lacking, demonstrating a key gap in the understanding of how they carry out their highly regulated functions within the confined space of the thick filament. Here we present the in situ structure of the thick filament in the bare zone, P zone and C zone in relaxed mouse cardiac myofibrils determined using our established workflow for focused-ion-beam milling and cryo-electron tomography (cryo-ET)[16,24,25]. The structure reveals the architecture of the thick filament, visualizing critical interactions between myosin, titin and MyBP-C, as well as MyBP-C links between thin and thick filaments.

## Structure of relaxed cardiac sarcomere

To obtain a near-native sample suitable for focused-ion-beam milling and cryo-ET analysis, we relaxed demembranated mouse cardiac myofibrils in the presence of EGTA, dextran and mavacamten (Methods). Mavacamten, a drug approved by the Food and Drug Administration to treat hypertrophic cardiomyopathy, stabilizes the OFF (relaxed) state of myosin[26].

The tomograms showed that the thick filaments were indeed in the OFF state, as defined by the absence of myosin binding to thin filaments (Fig. 1a). Using subtomogram averaging, we could determine the structure of the cardiac thin filament with tropomyosin in the blocked state (myosin-binding site occluded) at 8.2 Å resolution (Fig. 1b,d, Methods and Extended Data Figs. 1a, 2a,b and 3a,b) and the

structure of the relaxed thick filament (Fig. 1c,d, Methods and Extended Data Figs. 1b–j and 2c–k). To tease out greater detail in the analysis of the thick filament, we subdivided the filament into overlapping segments and determined their structures independently, obtaining subtle structural differences between the different segments. The resulting reconstructions ranged in resolution from 19.3 Å to 23.6 Å (Extended Data Fig. 2c–i).

Mapping of the structures into their determined $X$–$Y$–$Z$ positions within the reconstructed tomograms further enabled us to characterize the 3D arrangement of thin and thick filaments (Fig. 1b and Supplementary Video 1). In agreement with previous structural studies of vertebrate heart muscles, the myosin heads in our combined reconstruction are arranged in a quasihelical array with three-fold rotational symmetry[5–7] (Fig. 1c and Extended Data Fig. 4a–d). The combined reconstructions comprise the M band, P zone and C zone and reveal the positions of M-band proteins together with the first 31 myosin layers, encompassing 6 titin molecules and 9 cMyBP-C stripes (27 molecules of cMyBP-C; Fig. 1b,c and Extended Data Figs. 1b–j and 4a). This structure of the thick filament has a diameter of 32 nm, a length of 510 nm and a molecular weight of about 67 MDa and enables a description of the thick filament from the M band to the C zone in a region-by-region manner.

## Structures of myosin, titin and cMyBP-C

The segments containing cMyBP-C stripe numbers (nos.) 4–9 and the titin C-type super-repeats 3–8 have similar structures and were therefore averaged, yielding an improved resolution of 18 Å (Fig. 2a, Methods and Extended Data Fig. 2j). We then fitted atomic models of thick filament proteins (obtained either from experimental structures or AlphaFold2 predictions[27,28]) in the resulting reconstruction, allowing

us to assign all densities to the corresponding thick filament components (Fig. 2b and Supplementary Video 2).

The reconstruction revealed three distinct crown arrangements of myosin (defined as crowns 1, 2 and 3 from the M band to Z-disc direction) that form the outermost layer of the thick filament (Fig. 2d–i). Their double heads interact with each other within myosin dimers in a conformation known as the interacting heads motif[29] (IHM; Fig. 2c). In this OFF state, one of the heads ('blocked head') is folded back so that its actin-binding site is blocked by the other head ('free head'). This configuration has been suggested to correlate with the super-relaxed state[30,31], in which both motors have highly inhibited ATPase activity[29].

The myosin tails associated with the crowns form a large bundle of coiled coils at the centre of the filament (Fig. 2d–i). On the bundle of tails, there are three pairs of titin molecules that run parallel to each other but where each molecule (or chain) in the pair has a unique conformation. To uniquely identify the two chains, we named them titin-α and titin-β (Fig. 2a,b). Whereas the heads of crowns 1 and 3 do not interact with titin, the free head of crown 2 binds to the sixth domain of the titin C-type super-repeat (Fig. 2f,i), as can also be observed at cMyBP-C stripe No. 1 (Extended Data Fig. 5b,g–j).

The structure also demonstrates that cMyBP-C binds with its C-terminal domains C7 to C10 to an array of myosin tails (Fig. 3a–d and Extended Data Figs. 5b,d,f and 6b), thus providing visual confirmation of biochemical evidence that started to accumulate half a century ago[32,33].

Unexpectedly, the C-terminal region of cMyBP-C also interacts directly with the IHMs. Specifically, domain C10 binds to the motor domain of the free head of crown 3 while domain C8 forms broad electrostatic interactions with the motor domain of the free head of crown 1 (Fig. 3b,c and Extended Data Fig. 5b,c,e). These interactions of cMyBP-C with myosin motors suggest a direct role of cMyBP-C in the stabilization of the myosin OFF state in the C zone and explain why cMyBP-C can stabilize the super-relaxed state[34]. This highlights a paradigm shift for the role of cMyBP-C in sarcomere regulation and presents an interface at which pharmacological intervention for patients with hypertrophic cardiomyopathy can be explored.

## Structural arrangement of myosin tails

Although the organization of myosin tails has been determined in insects and spiders[35–37], so far we have no structural information to elucidate their organization in mammalian striated muscle, representing a gap in our understanding of the molecular mechanics of cardiac contraction. In the 43-nm repeat reconstruction of the C zone, only parts of the tails are visible, so to visualize their full extent we calculated a 200-nm-long helical extension starting from this reconstruction (Methods). The resulting 3D map allowed us to elucidate the tail arrangement associated with each of the three crowns (Fig. 4a,b). The tails of the three crowns interweave, interacting with each other, and form a compact rod. The tails of crown 3 form the core of the rod and interact with each other in their C-terminal section, suggesting that the myosin tails that are associated with crown 3 represent a possible nucleus for filament assembly.

To quantify and compare the curved nature of the tails, we calculated a sinusoidal compression percentage (Methods) that positively correlates to increases in tail curvature and resulted in 3.05%, 4.44% and 2.53%, for tails from crowns 1, 2 and 3, respectively. Notably, while the tails bear unique inflection points at different locations, crowns 1 and 2 share a strongly pronounced curve that correlates with a kink predicted by AlphaFold2 at the phenylalanine residue in position 1449 (Fig. 4a). This residue does not correspond to any skip residues predicted in the coiled-coil sequence and is not present in non-mammalian myosin II[38], possibly revealing a peculiar characteristic of the mammalian myosin II tail.

Within the C zone, the myosin tails have different degrees of interaction with titin. The tails associated with crown 3 have only minimal interaction, whereas those of crown 1 bind to titin-α, titin-β or both for their entire length (Extended Data Fig. 6a,c). Tails associated with crown 2 bind only to titin-β, mostly through their S2 domain (Extended Data Fig. 6b). Taken together, these data show that each crown within the cardiac thick filament exhibits a distinct arrangement of its tails and pattern of their interactions that are probably orchestrated and tuned by the titin chains, although the reverse process would also be conceivable. This suggests the existence of specialized roles for the myosins of each crown in length-dependent activation and contraction. The curved nature of the tails of crowns 1 and 2 and their direct interaction with titin support the observation that both titin chains not only act as counters of myosin molecules (as a molecular ruler), but can sense and relay muscle load, transducing a signal that might promote transition from the super-relaxed to the disordered relaxed state of the upstream crowns towards the M band.

## Variability across myosin crowns

To better understand whether there are differences between myosin heads along the filament, we analysed our combined reconstruction, for which we resolved 93 myosin double heads, spanning from the P-zone layers (P1, P2 and P3) to the C zone (from A1 to A28), corresponding to the titin region from the M band to the C-type super-repeat 2 (Fig. 5a–i). We observed that all myosin heads are in the IHM state, confirming that all 31 layers are in the OFF state. The pseudohelical arrangement of the 'crown 1, 2, 3' pattern, which we observed from cMyBP-C stripe No. 4 to stripe No. 9 is preserved throughout the C zone. However, in the regions closer to the M band there are noticeable structural differences (Fig. 5c–f and Extended Data Fig. 7b–d).

To quantify these variations, we calculated the orientations (Euler angles), azimuthal angle, axial positioning and radial distance of the IHMs for each crown (Extended Data Fig. 7a–d). The heterogeneity in the rise, twist, and angular orientation of myosin is particularly pronounced from crowns P1 to A9 whereas a more regular pattern emerges from A10 (Extended Data Fig. 7b,d). This is exemplified by crowns A5 and A8: although they localize in canonical 'crown 2' layers, these IHMs have a much wider radial distance that makes them the closest to the thin filaments after P1 (Extended Data Fig. 7d). Moreover, their free heads are projected further outwards (Fig. 5b–f), as documented by their strongly negative $\beta$-angle (Extended Data Fig. 7b,c). This variability suggests that these crowns might have unique capacities in terms of strain susceptibility and activation rate. In particular, as they are stabilized only by a weak interaction with titin, crown 2 seems likely to be the first responder in length-dependent activation, with exceptional layers such as A5 and A8 being the most representative cases.

Although all myosins in our reconstruction are in the IHM state, their stabilization varies and relies on a variety of interactions, depending on their specific molecular context, that involve binding sites on myosin S2, titin and cMyBP-C (see also below). Conceivably, this organization along the filament is indicative of an ability for the myosin heads within the thick filament to be fine-tuned by regulatory mechanisms of muscle contraction within the thick filament in adaptation to specific physiological conditions[39].

## Titin organization in the P and C zones

Despite the essential role of titin in sarcomere assembly and function, its detailed structural arrangement has remained hitherto unknown[1]. We used our combined reconstruction, spanning from the P-zone layers (P1, P2 and P3) to the C zone (from A1 to A28), to build an atomic model of titin. To this end, we used AlphaFold2 predictions of titin that unequivocally dictated the register and the position of

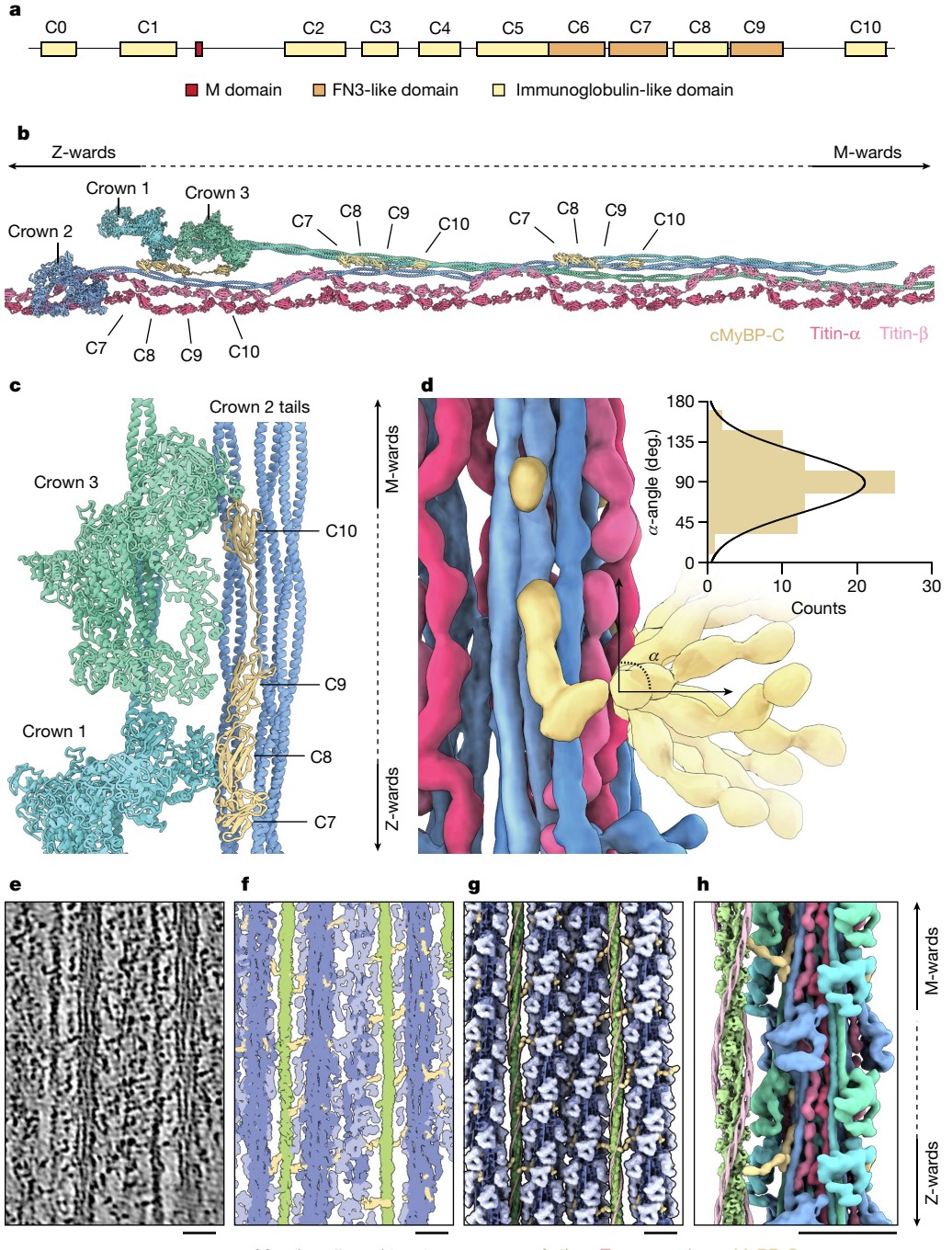

**Fig. 3 | cMyBP-C forms links between the thick and thin filaments. a**, Domain map of cMyBP-C (1,270 amino acids). The C-terminal domains from C7 to C10 bind to the thick filament, the central domains C3 to C6 form the bridge, and the N-terminal domains bind to the thin filament. **b**, Structural model depicting the three binding regions of cMyBP-C on the tail of crown 2. **c**, The epitope for the binding of the C-terminal domain of cMyBP-C to the thick filament is formed by an array of crown 2 tails and places C10 and C8 in contact with crowns 3 and 1, respectively. **d**, Schematic representation of the bridging region of cMyBP-C. Flexibility is shown together with a plot of the angular distribution of the bridge measured in the tomographic volumes. **e**, Tomographic slice (0.94 nm thickness) of the C zone in the relaxed state. **f**, Tomogram segmentation of a 9.4-nm slab from the same region as in **e**, depicting the thin filament (light green), the thick filament core (blue), the myosin heads (light blue) and cMyBP-C (yellow). **g**, A 3D model of the sarcomere organization in the same slab as in **e**, showing the flexible cMyBP-C links, spanning 3–4 globular domains, linking thick and thin filaments. **h**, Close-up view of the connection of two consecutive cMyBP-C proteins to the same thin filament. Scale bars, 20 nm.

titin domains. This led us to describe the molecular organization of titin from the C zone (specifically from C-type super-repeat 2) to the M band (Fig. 5a–c,f,g,i,j) and reveal its interplay with the other thick filament components, providing molecular information useful for the understanding of both sarcomere function and causative effects of disease.

Notably, whereas the three titin-α chains run along the entire length of the thick filament and reach to the P zone of the opposite half-sarcomere, titin-β is interrupted abruptly at myosin crown A1 (Fig. 5c,f,g,i,j and Extended Data Fig. 8). Thus, there are six chains in the C zone, but only three in the P zone. Previous antibody labelling studies determined that titin C-terminal domains span across the M

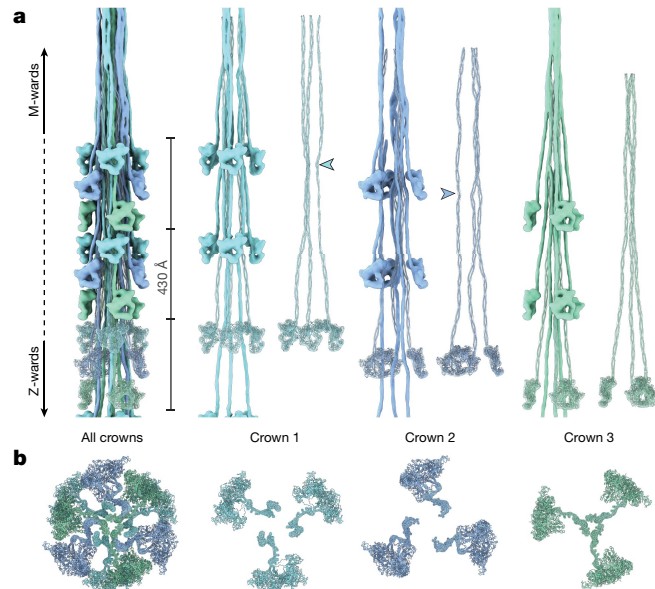

**Fig. 4 | The spatial arrangement of myosin within the C zone of the thick filament. a,b,** Side (**a**) and Z-ward (**b**) views of a combined map and atomic models of a section of the thick filament depicting only myosins. The myosins follow a three-fold rotational symmetry and a pseudohelical symmetry with a 43-nm rise and 0° twist. The asymmetric unit comprises three myosin crowns, each composed of three myosin double heads. The tails form the backbone of the thick filament. The tails of crown 2 represent the outermost layer, followed by those of crown 1 and crown 3. Whereas the tails of crown 3 run relatively straight, the tails of crowns 1 and 2 exhibit increased degrees of undulation. The arrowheads indicate an inflection point, common to the tails of crowns 1 and 2, which could also be directly observed from the AlphaFold2 predictions of the myosin tails at the phenylalanine residue in position 1449.

band, with m9–10 localizing proximal to P1' (ref. 40). Consequently, only three chains enter the M band from each side of the thick filament, resulting in only six chains of titin in the M band instead of twelve.

In the C-type super-repeats, titin-α and titin-β run alongside each other with only sparse and presumably weak interactions, resulting from the molecular contacts of titin with myosin tails. In both the P and C zones, titin chains show a spring-like arrangement, with recurring curved indentations following a 3-4-4 pattern that matches the alternating FN3 and immunoglobulin-like sequences found in the super-repeats of titin[41] (Fig. 5a,c,f,g,i,j). Beside this common motif, we observed a certain degree of variability between different C-type super-repeats. These variations are mostly accounted for by the disorganized linkers scattered between domains all along the structure and are more pronounced on titin-β. Super-repeat domains 4 to 8 have the highest conformational variability in titin-β. These domains interact with the S2 domain of crown 2 (Extended Data Fig. 5b,g–j), and their variability is reflected in the heterogeneous configuration of crown 2 IHMs (Extended Data Fig. 7b–d) as already discussed.

In the P zone, which is the transition zone between the A and M bands, the pattern of the titin C-type super-repeat is replaced by seven domains (from A164 to A170) that lead to the titin kinase (TK) domain (Fig. 5a–g,i,j and Extended Data Fig. 8b,c). Following the titin-α chain in our reconstruction, we identified a bulkier region behind the P1 crown that fits the size and shape but not the expected position of the TK domain (Extended Data Fig. 8b,c). As this position disagrees with that previously assigned at about 52 nm from the M1 line by immuno-electron microscopy using a monoclonal immunoglobulin M (IgM) anti-TK domain antibody[40], we generated a new affinity-purified TK domain antibody and used it to ascertain the TK domain position

by super-resolution microscopy in mouse cardiac and mouse and rabbit psoas myofibrils (Extended Data Fig. 8f). The position of the TK domain thus determined was 78 nm ± 8 nm from the M1 line. This is most consistent with the TK domain density at the P1 crown but, given the expected resolution limits of stimulated emission depletion microscopy, does not rule out localization of the TK domain at both crowns A1 (see below) and P1 (43 nm apart).

The TK domain is closely connected via its C-terminal regulatory tail to the adjacent m1 domain, as has been previously proposed[42,43]. Notably, the titin m1 domains interact with the S2 domains of the myosins in crown P2, while the subsequent titin m2 and m3 domains are wrapped around the myosin tails associated with the P1 crown (Extended Data Fig. 8b,c).

In the C zone, the titin-β chain follows the same register of titin-α; however, after domain A154, in proximity of crown A2, its organization changes to a bow-tie-shaped structure and its chain is absent after crown A1 (Extended Data Fig. 8d,e). On top of the A1 crown, we identified another bulky density reminiscent of the TK–m1 configuration. However, the limited resolution does not allow clear identification of the domains. We propose that the bow-tie-shaped structure might correspond to 13 titin domains (from A158 to A170), connecting the rest of titin-β to the β-TK–m1 domains (Extended Data Fig. 8d,e) and that titin-β terminates its run along the thick filament on the A1 motor domains, perhaps after proteolytic cleavage by calpain 3 (CAPN3) downstream of titin m1 (ref. 44). Thus, the TK domains of titin-α and possibly of titin-β would localize at 77 and 105 nm from the centre of the M band (in contact with crowns P1 and A1, respectively), interacting with their S1 and S2 domains in privileged positions that could couple myosin head activation to mechanosensing by the TK domain.

## Molecular topology of the cardiac M band

The M band, comprising six transverse 'lines' (M1–M6), is a complex network of structural proteins, metabolic enzymes and proteostatic machinery components[1,3]. Proteins in this network are reported to anchor the M region to the sarcolemma, to stabilize the myosin filament hexagonal lattice and to serve as a signalling scaffold that integrates mechanical forces, energy balance and protein turnover[1,3]. Our reconstruction of the thick-filament M region shows a two-fold mirror axis centred around the M1 line, perpendicular to a three-fold rotational axis. We could clearly identify the myosin crowns in the P zone and also follow the tails from the four innermost crowns, which run antiparallel to each other, thus confirming previous indications[4] that the P1 tails terminate in the proximity of the P1 crown from the opposite polarity (P1'; Fig. 6a,c).

In addition to myosin, we observed several densities protruding from the core of the tails, reflecting proteins and domains specific to the M band such as myomesin 1 and 2 (ref. 3), obscurin-like-1 (OBSL1)[45], C-terminal domains of titin[40] and accessory proteins such as muscle-type creatine kinase (M-CK). We identified a density spanning from the M lines M4' to M4 that binds to the thick filament core at 85 Å and 230 Å and projects outwards with a three-fold symmetry into the hexagonal lattice. Taking into consideration its localization, arrangement and composition from about 4-nm-long domains, we suggest that this density represents a complex of myomesin 1, OBSL1 and possibly myomesin 2 (Fig. 6a–c). Unexpectedly, in our relaxed cardiac M-band reconstruction, we observe densities at 118 Å from M1, protruding towards neighbouring thick filaments in the lattice (Fig. 6c). However, the expected cross-connecting M bridges, as they would be inferred from the dimeric myomesin 1 structure[46], are averaged out during the refinement owing to their flexibility.

We could not determine the location of obscurin. This is probably due to a combination of structural flexibility and the localization of obscurin at the myofibril periphery, as opposed to the central

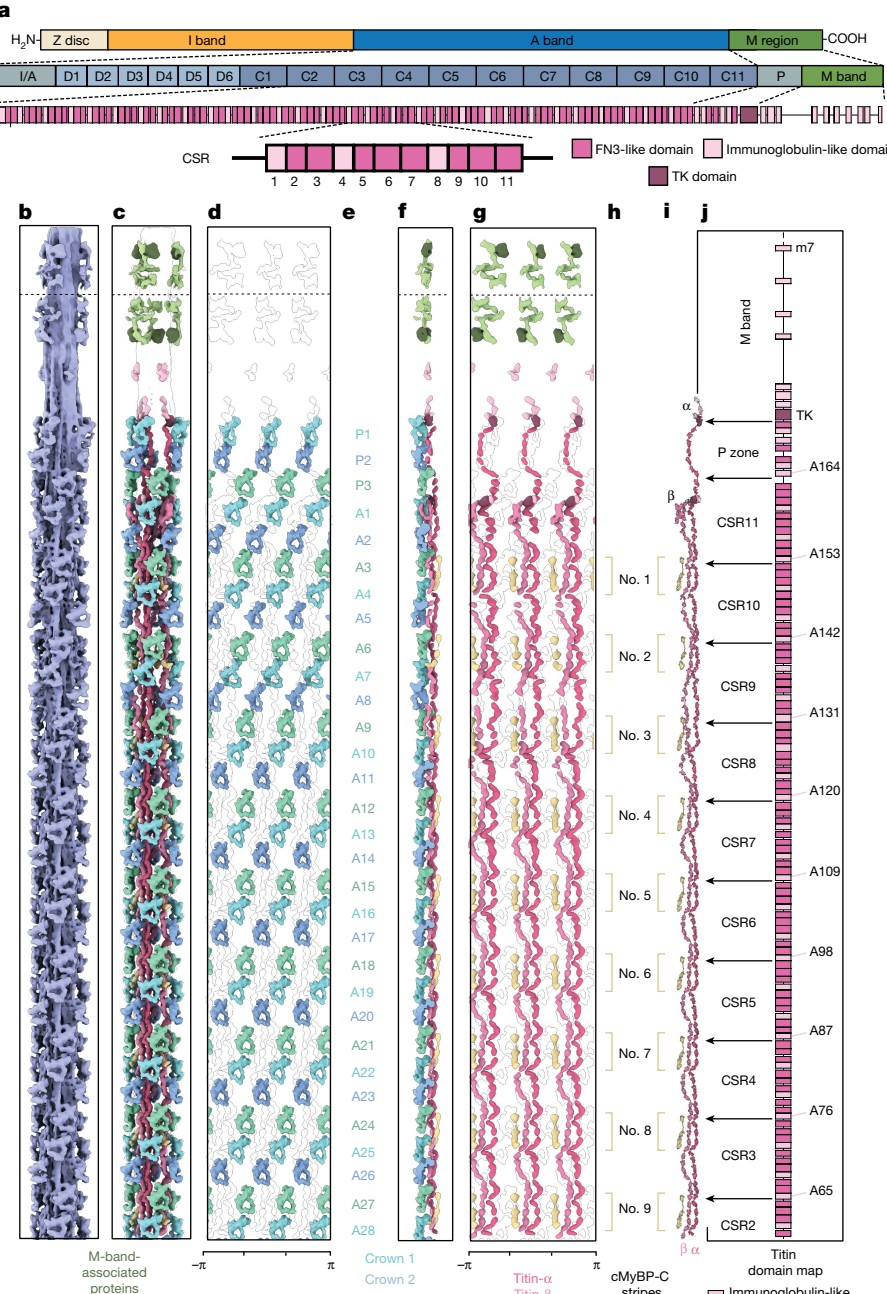

**Fig. 5 | Arrangement of myosin heads, cMyBP-C and titin from the M band to the C zone. a**, Domain map of titin, showing its highly modular organization. The distal region of the A band comprises six 7× D-type super-repeats, and the central region contains eleven 11× C-type super-repeats (CSRs). The proximal region (P zone) contains the TK domain, and the M-band region presents ten immunoglobulin-like domains connected by seven disordered interspacing regions, m1–10 and is1–7, respectively. **b**, Cryo-ET map of the thick filament, spanning from the M4′ line to the C zone. **c**, Cryo-ET map of the thick filament in which myosin tails have been removed for visual clarity. **d**, 'Unrolled' depiction of the filament, revealing the spatial organization of the myosin heads. Position and arrangement of myosin heads from the P zone (crowns P1, P2 and P3) to stripe no. 9 of cMyBP-C (from crown A1 to A28). **e**, Designation of the crowns. **f**, Side view of the 'unrolled' filament with the centre of the thick filament to the right side. **g**, 'Unrolled' depiction of the filament, revealing the spatial organization of the 6 titin and 27 cMyBP-C molecules. **h**, The structural arrangement reveals the nine stripes of the C-terminal domain of cMyBP-C, anchored on the thick filament in proximity to the interface of different C-type super-repeats. The numbering and the positions of the cMyBP-C stripes are indicated. **i**, Atomic models of a single asymmetric unit, depicting titin-α and titin-β together with the nine cMyBP-C stripes. **j**, Domain map of titin showing the blueprint of titin from the M band to C-type super-repeat 2 and its correlation to our reconstructions.

sarcomeric localization of OBSL1 (ref. 45). As described above, only titin-α reaches the M-band region with its C terminus. It consists of the TK domain, ten m domains (m1–10) and seven interspacing regions (is1–7)[40]. Within each asymmetrical unit of the bare zone, at 427 Å from the high-density M line M1, we observed a density that fits the size and shape of two immunoglobulin-like domains, possibly the adjacent m8

and m9 titin domains. In addition, there are other small densities that might correspond to m1, m2, m3 and m10. The interspacing regions are very thin and do not contain predicted secondary structure elements, and therefore cannot be seen in our reconstruction. In line with previous suggestions[40], these observations support the idea that the titin-α M region spans across the M band. The identification of M-line densities

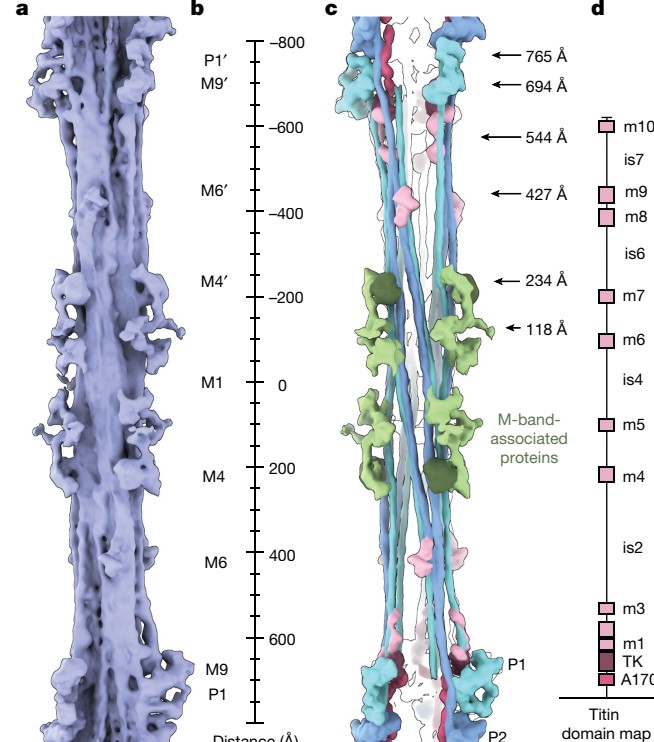

**Fig. 6 | The layout of the M band. a**, Density of the thick filament in the M-band region. The overall structure shows a two-fold rotational axis perpendicular to a three-fold rotational axis ($D_3$ symmetry). **b**, Ruler, providing markers for the expected high-density M lines in the cardiac sarcomere (M1, M4, M6 and M9). M1 is absent in our reconstruction, probably owing to the lattice organization being averaged out during the refinement process. **c**, The most clearly identifiable densities from the M-band map are the tails of the first two myosin crowns, P1 and P2, together with an unresolved cluster of M-band-associated proteins. **d**, Domain map of the C-terminal region of titin, which consists of the TK domain, the ten m domains (m1–10) and seven interspacing regions (is1–7). The length of the interspacing regions roughly reflects the length of the extended polypeptide chains predicted from AlphaFold2 in those regions.

in their near-native state provides a blueprint for titin m-domain localization and structural rationalization of reported biochemical interactions[3,40,45,47,48]. Detailed assignment of the densities will require a combination of site-directed labelling of the domains and cryo-ET in the future.

Nonetheless, the structure of the native cardiac M band presented here in situ represents an initial model of the M band at the molecular scale. The crosslinks formed by M-band proteins are dictated by the three-fold rotational symmetry of the thick filament. However, the interactions of titin with OBSL1, and OBSL1 with myomesin 1, are all in a 1:1 complex[48], posing a symmetry mismatch between these constituents and the previously expected six titins from each half-sarcomere entering the M band. Our analysis resolved this paradox, showing that the full M-band region of titin is present only in three copies in each half-sarcomere, directing the assembly of the six observable M-links (three in each half-sarcomere).

## cMyBP-C links thick and thin filaments

In our reconstruction of the thick filament, we found that the C7–C10 domains of cMyBP-C bind to an array of three consecutive tails (Fig. 3b,c) associated with crown 2 in nine regions of the C zone[18,49] (Fig. 5f–j). C7–C10 run longitudinally along the filament: for each cMyBP-C, C10 is in the M-band direction and C7 is oriented towards

the Z disc, spanning a 16-nm region at the interface between titin C-type super-repeats (Fig. 5f–j). Initial biochemical data suggested that the confinement of cMyBP-C to the C zone might depend on a direct interaction between C7–C10 of cMyBP-C and the titin C-type super-repeats[11]. We do not observe any direct interactions between the two proteins in the relaxed state. Instead, we clarified that the C-terminal domains bind to a specific configuration of myosin tails that emerges in the C zone as a complex interplay between myosin tails and titin C-type super-repeats (Fig. 3b–d and Extended Data Fig. 5b–d,f). Specifically, C7–C10 bind to regions that emerge from an array of tails associated with crown 2 belonging to three consecutive C zones (Fig. 3b,c) via strong electrostatic interactions (Extended Data Fig. 5c,f). This configuration also resolves the absence of cMyBP-C in the titin region of titin C-type super-repeat 1, as the site where cMyBP-C would bind is in this case composed of tails of the D-type super-repeat 6, which we have not resolved in our structure, but which probably have a different conformation.

In the C-terminal region of cMyBP-C, C7 is the last clearly resolved domain and seems to act as a pivot point for the remaining domains that reach away from the thick filament (Fig. 2a,b,e). The bridging region of the protein indeed showed great flexibility and had only weak densities (Fig. 3d and Extended Data Fig. 9a). Although stripe nos. 1 and 2 showed biases towards a specific lattice organization, the cross-sectional analysis at the different stripes revealed that the cMyBP-C N-terminal region can bind to either of the two neighbouring thin filaments (Extended Data Fig. 9a). Remarkably, cMyBP-C stripe no. 2 shows some unique features (Extended Data Fig. 10b,c): the free head of crown A7 is not stabilized by an interaction with cMyBP-C C8, as is observed in all of the other stripes; instead its IHM is stabilized by interactions with myosin light chains of the neighbouring crowns.

To validate this pliability of cMyBP-C links, we back-plotted the structures into the reconstructions of the tomographic volume and manually traced the flexible globular domains connecting thick and thin filaments (Fig. 3e–h, Methods and Supplementary Video 3). We observed the presence of several cMyBP-C links in the C zone, suggesting that the N-terminal region can indeed interact with the thin filament. Despite our efforts, subtomogram averaging did not resolve any clear arrangement of the N-terminal region on the thin filament, indicating a high heterogeneity in these interactions, in agreement with previous reconstructions from isolated components[50]. Our segmentation exposed the presence of a tether formed by 3 to 4 domains that blend in the signal of the thin filament with 43-nm intervals and high degrees of flexibility (Fig. 3e–h). Thus, the flexibility of the links can probably accomodate the mismatch of the 37-nm actin repeat in thin filaments and the 43-nm cMyBP-C repeat in thick filaments. Of note, the thin filament with bound N-terminal cMyBP-C domains appears in the OFF state, suggesting that cMyBP-C, at physiological phosphorylation levels and in the absence of myosin cross-bridges, is not a thin filament activator.

cMyBP-C represents a crucial site of cardiomyopathies and has thus far remained an enigmatic component of the sarcomere, with its function in modulating myosin activation being proposed but not well understood. Collectively, our analysis of cMyBP-C suggests multiple roles in sarcomeric mechanosignalling. On the one hand, it is reasonable to speculate that under thick filament strain, the sinusoidal path of crown 2 tails can experience a minimal but significant stretch, resulting in the alteration of the binding sites for the C-terminal region of cMyBP-C and perturbing its interactions with the free heads. As such, the interactions of the C-terminal domains with the IHMs of crown 3 and 1 would represent a transducer for tension and stretch, independently of thin filament positioning. On the other hand, the crosslinks to the thin filament can sense the stretch on and sliding of the thin filament and relay the tension to both the C-terminal and N-terminal domains of cMyBP-C, while domain C1 and C0 might extend further to bind the myosin light chains on the thick filament surface, either in *cis* or *trans*.

## Conclusions

Our study provides insights into the fundamental organization and regulation of cardiac thick filaments in vertebrates. Our cryo-ET structure of the sarcomere in the relaxed state together with our previously determined structures of the sarcomere in the rigor state[16,25] provide insights into muscle architecture and its molecular complexity, intricacy of regulation and sophistication. The structure of the native thick filament provides a 3D representation of how the myosin heads and tails, MyBP-C and titin are organized in different areas of the filament, explaining how these components can interact during muscle contraction. It also reveals distinct differences in how certain domains of the sarcomere are organized when compared to results obtained using traditional reconstituted approaches and classical electron microscopy methods.

Our analysis revealed a plethora of conformations and stabilization mechanisms for the IHM, which are region dependent and rely on various interactions with the m1 domain of titin, the C-terminal region of cMyBP-C, super-repeats of titin, other myosin tails and myosin light chains. These findings support a model of a thick filament that is capable of sensing, integrating and modulating responses to numerous signals such as β-adrenergic-dependent phosphorylation, Ca²⁺ concentration, length and mechanical load. In addition, we revealed the organization of titin along the thick filament and the genuine position of its TK domain not in the M band but juxtaposed at the interface between the M band and the A band. Such a localization indicates that the role of TK in regulating the sarcomere may be more tightly coupled to the state of the myosin heads than previously thought.

The six titin chains probably influence the arrangement and pattern of interactions of each myosin tail. Thus, it is understandable how titin acts as a scaffold and a molecular ruler for myosin assembly during sarcomerogenesis, as has been previously suggested[51]. This is an interesting reflection of the protein nebulin found in the thin filament[25]. Furthermore, the close interaction and spring-like appearance of both titin and the myosin tails and their connection to myosin heads make a compelling case for a load-dependent activation mechanism of the thick filament promoting the transition from the super-relaxed to the disordered relaxed state.

We were able to observe biologically important structural variability along the thick filament, which is a crucial aspect needed for understanding sarcomere function and regulation. Static structures in isolation tell us little about the interplay and dynamics of multiple, highly regulated myosin heads interlinked to one another. These important details would have been lost with most structural biology methods and demonstrate the merits of cryo-ET in visualizing this variability in the native state. The cMyBP-C links connecting thin and thick filaments are another example of the power of cryo-ET. Finally, the insights gleaned through our thick filament structures provide a framework for understanding muscle development, function and disorders.

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

## Methods

### Myofibril preparation and vitrification

Demembranated left-ventricular mouse myofibrils were prepared as previously described[25]. The myofibrils were collected by centrifugation at 3,000*g* for 2 min at 4 °C, followed by two washes with pre-relaxing buffer (100 mM TES pH 7.1, 70 mM KCl, 10 mM reduced gluthatione, 7 mM $MgCl_2$, 25 mM EGTA, 20 μM mavacamten, 5% dextran T500). To prepare for plunging, the pre-relaxing buffer was replaced with final relaxing buffer (pre-relaxing buffer plus 5.5 mM ATP). The relaxed myofibrils were then frozen onto Quantifoil Au R2/2 $SiO_2$ 200 mesh grids using a Vitrobot (Thermo Fisher Scientific). The myofibril suspension was incubated on the grid at 25 °C and 100% humidity for 30 s, blotted for 30 s from the opposite side of the carbon layer, and plunged into a liquid ethane–propane mixture.

### Cryo-focused-ion-beam milling and cryo-ET

The preparation of lamellae for cryo-ET data acquisition was carried out by cryo-focused-ion-beam (cryo-FIB) milling using an Aquilos 2 cryo-FIB scanning electron microscopy system with a cryo-shield, according to previously described protocols[16,24,25] aiming for lamellae with a final thickness of 180 nm (range from 90 to 250 nm). The data acquisition was carried out with a Titan Krios transmission electron microscope (Thermo Fisher Scientific), fitted with a K3 detector and an energy filter (Gatan). The acquisition of overview images of myofibrils in the lamellae was carried out at a nominal magnification of ×6,700 to identify the C-zone regions.

The tilt series were acquired targeting the C zones at ×81,000 nominal magnification. The pixel size was calibrated to 1.146 Å using the 143.3 Å peak in the fast Fourier transform of the final thick filament reconstruction (from the M band to the C zone; Extended Data Fig. 4a,c). A dose-symmetric tilting scheme[52] was applied during acquisition with a tilt range of −50° to 50° relative to the lamella plane at 2.5° increments. The sample was subjected to a total dose of 120 to 160 electrons per square ångström. Tilt series were acquired using a defocus between −3 and −6 μm. All images and a total of 89 tomograms were acquired using SerialEM[53].

### Tomogram reconstruction and particle picking

Motion correction and contrast transfer function estimation were carried out in Warp[54]; tilt series alignment was carried out in IMOD[55]. Final tomogram reconstruction and subtomogram extraction were carried out in Warp. After binning the tomograms to a pixel size of 0.92 nm and low-pass filtering them at 60 Å, we used SPHIRE-crYOLO[56] to pick and trace both the thick and the thin filaments (Extended Data Fig. 1a,b). The average sarcomere length of the dataset was 2.326 μm (s.d. = 0.11 μm, *N* = 45), indicating that the sarcomeres are not hypercontracted. This is also supported by the 1.95:1 (102 thin filaments, 53 thick filaments) thin/thick filament ratio in the two tomograms we segmented.

### Thin filament processing pipeline, model building and visualization

The traced thin filaments were resampled with an intersegment distance of 18 Å, leading to the extraction of 365,971 subtomograms with a box size of 293.5 Å (128 pixels, binning 2). Using customized scripts, each subtomogram was rotated to orient the thin filaments parallel to the *X*–*Y* plane using the previous angles from the tracing. Then, the central slab of 100 slices was projected and used as an input for 2D classification with ISAC[57–59]. The classes that did not show a clear presence of thin filaments were discarded and the remaining segments were re-extracted as subtomograms and processed in RELION 3.1 (refs. 60,61) via gold-standard dataset splitting. The initial helical reconstruction led to a 14.3-Å-resolution map (0.143 FSC criterion) with 27.4 Å rise and −167.2° twist (Extended Data Fig. 1). After removing the duplicated particles using a customized script, 100,447 subtomograms were further refined with two different masks that either covered the entire density of the thin filament, or included only the F-actin density. The full thin filament (F-actin and tropomyosin) was refined with helical reconstruction and reached a resolution of 8.2 Å while the refinement of F-actin alone resulted in an 8.3-Å-resolution map. For the thin filament map and the F-actin map, we applied a B-factor of −200 and −100, respectively. The two maps were aligned in ChimeraX[62] and the individual chains from Protein Data Bank model 6KN7 (ref. 63) were placed in the density with rigid body fitting. The model of the coiled coils of tropomyosin was improved with Namdinator, using automatic molecular dynamic flexible fitting[64]. The final composite map (Extended Data Fig. 3a) was created by combining F-actin from the reconstruction of F-actin alone and tropomyosin from the full thin filament reconstruction using the color zone and splitbyzone functions in ChimeraX.

### Thick filament processing pipeline

The traced thick filaments were resampled with an intersegment distance of 130 Å, leading to the extraction of 67,492 subtomograms with a box size of 1,280 Å (160 pixels, pixel size 8 Å). 2D classification was carried out similarly to the thin filament processing, using a central slab of 400 Å, resulting in 37,118 high-quality particles that were re-extracted as subtomograms. 3D classification with refinement and helical reconstruction (430 Å, 0° twist) resolved four classes that showed different orientation of crown 2. Class A showed 'projected' IHMs, class B had a mixture of conformations resulting in fuzzy density for crown 2 IHMs, and class C showed 'retracted' IHMs (Extended Data Fig. 1b). The refined coordinates were used to individually re-extract the three classes from Warp, using a box size of 144 pixels and a pixel size of 4 Å. The individual classes were refined in RELION 3.1 via gold-standard dataset splitting using a featureless cylinder as an initial reference and their coordinates were mapped back into the tomograms using ArtiaX[65]. Classes B and C showed particles organized in filaments but randomly distributed away from the M line. Class A, which was later resolved as the segment from crown A8 to A12 (cMyBP-C stripe No. 2; Extended Data Fig. 1g), showed a unique distribution within the sarcomere, localizing as an array of particles parallel to the M line, roughly 200 nm away from it. We later understood that the crowns 2 contributing to the helical average of class A were crowns A5 and A8. As it turned out, whereas all crowns 2 have a *β*-angle of about +30°, crowns A5 and A8 have a *β*-angle of about −25°, explaining why the 'projected' class stood out during the refinement with imposed helical symmetry (Extended Data Fig. 7b,c). As we could identify the location of class A along the thick filaments, we wrote a customized script to calculate the 'axially shifted coordinates': starting from a known anchor point, we calculated the 3D coordinates of the next thick filament segments, shifting the position of class A coordinates 43 nm Z-wards and 43 nm M-wards (Extended Data Fig. 1f,h). This allowed us to resolve distinct reconstructions for each thick filament segment. With this strategy, we progressively resolved eight structures of the thick filament, spanning from the M band to titin C-type super-repeat 2, in the C zone (Extended Data Fig. 1). The resulting 3D maps showed high variability in resolution within different regions of the same map leading to oversharpening of the more flexible regions; we therefore used LocSpiral[66] to improve map interpretability. All reconstructions showed a three-fold rotational axis and were therefore refined with $C_3$ symmetry. The M-band reconstruction further revealed two orthogonal two-fold rotational symmetry axes that intersect at the three-fold axis at an angle of 60° and was later refined applying $D_3$ symmetry.

To build the final composite map, the power spectra of the reconstructions were normalized with relion_image_handler[61] and a soft cylindrical mask of 15 px (about 60 Å) was applied to all maps. The filtered reconstructions were aligned in ChimeraX (fit in map) and the final map was created by merging the densities of the different segment, using the maximum value at each voxel (volume add, volume maximum in ChimeraX). To obtain a continuous and homogeneous density that we

could use to trace the myosin tails, we took our 18-Å reconstruction of the last five stripes of cMyBP-C and extrapolated a 200-nm-long helix with relion_image_handler (430 Å, 0° twist)[61].

## Model building and visualization of the thick filament

The model of the thick filament was built using a combination of previously available models and AlphaFold2 predictions[28]. For the model of myosin II, we started from the IHM of human β-cardiac heavy meromyosin (Protein Data Bank entry 5TBY; ref. 27) while the tails were predicted in AlphaFold2 using the amino acid sequence of MYH7 from *Mus musculus* (5 segments of about 250 amino acids with about 20 amino acids overlap). Similarly, the C-terminal domain of cMyBP-C was predicted in AlphaFold2 using the last 590 amino acids of MYBPC3 from *M. musculus*. The region of titin from domain A101 to m3 (amino acids 24,760–32,350) was submitted for prediction as multiple entries (each ≈950 amino acids) with overlapping terminal domains. AlphaFold2 predictions of titin resulted in distinctive structural motives (for example, TK–m1, C-type super-repeat domains 1 to 3, C-type super-repeat domains 7 to 9 and cMyBP-C C8–C9) that unequivocally dictated the register and the position of titin domains. The model culminated in titin domains and the 9 cMyBP-C stripes with distances from the M line in agreement with previously published data from immunoelectron microscopy (for example, titin domains A77–78, A80–82, A153, A165 and cMYBP-C C7 domain)[15,18]. The models were initially built in the map using rigid body fitting and their final organization was later adjusted with multiple rounds of molecular dynamic flexible fitting in Namdinator[64], starting with 40-Å low-pass-filtered densities and gradually using the higher-resolution maps. The models spanning from crown A18 to A28 were obtained by cloning the model spanning from crown A15 to A17. For the visualization in Fig. 2a, we used the ChimeraX functions colorbyzone, splitbyzone and Gaussian filter with standard deviation = 3. The depictions in Fig. 5c,e,f were obtained using the Chimera unroll function on the structure of all components individually. ChimeraX lighting was set as follows: soft intensity 0.1; direction 0.577, −0.577, −0.577; color 100, 100, 100; fillIntensity 0.5; fillDirection −0.81, −1,1; fillColor 100,100,100; ambientIntensity 1.4; ambientColor 100,100,100; shadow 1; qualityOfShadows finer; depthBias 0.01; multiShadow 64; msMapSize 2000; msDepthBias 0.004; moveWithCamera 1; depthCue 0. ChimeraX cartoon style was set as follows: width 2.5; thick 1; xsection oval strand width 3.2; xsection rect coil width 3.2; thickness 1.2. ChimeraX colour palette was as follows: thick filament core #707ec1; crown 1 #7ed5dc; crown 2 #7ea6dc; crown 3 #7edcb4; essential light chain #e154d8; regulatory light chain #9a41de; myosin blocked head #55667e; myosin free head #ace2ff; titin-α #d95d87; titin-β #dc7ea6; TK #844b63; m1-9 #eab1c9; M-band proteins-A #546845; M-band proteins-B #a8d18a; cMyBP-C #dec98f; F-actin #a8d18a; tropomyosin A #d1a8a8; tropomyosin B #d1a8bd; troponin #a8a8d1.

## Position and orientation of myosin crowns

To quantitatively describe the 3D arrangement of each crown, the thick filament was modelled as a cylinder, and the myosin IHMs were represented as triangles. The vertices of the triangles were determined by the coordinates of three points: the ATP-binding site in the free head, the same site in the blocked head, and the head–tail junction site. For each IHM, the Euler angles are calculated in a 3D Euclidean space with the origin on the centroid of the triangle, the x axis parallel tangential to the cylinder, the y axis parallel to the radius, and the z axis parallel to the axis of the cylinder. The coordinate system was calculated for each IHM to obtain the Euler angles ($\alpha$, $\beta$ and $\gamma$) specific to each crown (Extended Data Fig. 7a). To obtain azimuthal angle, radius and z-axis height, we used the centroid of each IHM, calculated the cylindrical coordinate and inferred twist, radial distance and rise, respectively (Extended Data Fig. 7d).

## Sinusoidal compression percentage

To quantify the curviness of the myosin tails, for each tail, we first obtained the atomic coordinates of the α-carbons for the two amino acid chains in the coiled coils. We then traced a new 3D curve running through the central points between each α-carbon couple. For each curve segment, we calculated the sinuosity S by the ratio of the length of the curve C to the Euclidean distance between the ends L:

$$S = \frac{C}{L}$$

The sinusoidal compression percentage (SCP) is then given by:

$$SCP = (S - 1) \times 100$$

## Tomogram segmentation and cMyBP-C links

To describe the 3D organization of the sarcomere components, we selected two representative tomograms (Figs. 1a and 4a and Supplementary Videos 1 and 2) and denoised them using cryo-CARE[67]. With a customized script, we mapped back each subtomogram using a binary mask of their corresponding structure, matching the coordinates and orientations obtained from the 3D refinement. The resulting binary MRC files were imported in Dragonfly[68] and used as a template for pseudo-segmentation of the tomograms. The resulting label layers were manually validated by inspecting each tomographic slice for unassigned densities and further tracing the flexible components that were averaged out during the refinement (that is, the cMyBP-C links from thick to thin filament). After clearly identifying and segmenting 76 cMyBP-C links in our tomograms, we measured the angle that the link formed relative to the thick filament z axis, using the position of the C7 domain as a pivot point. The angular distribution was plotted in GraphPad Prism.

## Antigen expression and purification

A fragment of the TK domain encompassing human TTN transcript variant-IC (NM_001267550.1), residues 33812–34076, was expressed in *Escherichia coli* BL21 [DE3] cells in fusion with an N-terminal His₆ tag. The insoluble fragment was extracted from inclusion bodies with 8 M urea, 50 mM potassium phosphate pH 8.0, 0.5% Tween 20 (buffer B) by sonication with a Branson sonifier microtip on ice. Insoluble material was pelleted at 15,000 r.p.m. for 20 min in an SA 600 rotor (Sorvall) and the soluble supernatant was applied to an Ni-NTA column equilibrated in buffer B. After washing the column as above in buffer B, bound protein was eluted with 250 mM imidazole in buffer B and equilibrated stepwise against 6, 4, 2 and 0 M urea in 40 mM HEPES buffer pH 7, 50 mM NaCl, 4 mM dithiothreitol and 0.1% Tween 20 (buffer C). The insoluble precipitate was spun down and the soluble protein was further purified by gel filtration purification on a Pharmacia Superose 12 column equilibrated in buffer C. The purified kinase fragment was used for commercial rabbit immunization, and serum was collected after three booster injections.

## Cloning, expression and purification of rat titin A170-kinase

For affinity purification, a soluble TK construct, A170-kinase, was used. The sequence encompassing the A170 (FN3) and kinase domains of rat titin (XM_008775521.1 residues 31897–32344) was cloned into a modified pCDFDuet vector containing an N-terminal His-tag, expressed in *E. coli* strain BL21 [DE3] using standard protocols and purified by nickel affinity and size-exclusion chromatography according to ref. 69.

## Antigen coupling and affinity purification of anti-TK antibody

A 1 mg quantity of purified A170-kinase was dialysed into coupling buffer (100 mM sodium phosphate pH 8, 250 mM NaCl, 1 mM dithiothreitol), and then coupled to 2 ml NHS-activated Sepharose 4 Fast Flow slurry following the manufacturer's instructions (Cytiva Life Sciences). Antibody affinity purification was carried out using standard procedures described previously[70]. Following equilibration with

10 ml PBS with 0.05% Tween 20, 5 ml of the rabbit anti-kinase serum was applied to the A170-kinase–Sepharose column, which was then washed with 20 ml PBS containing 0.05% Tween 20, 4 ml PBS and finally 4 ml 50 mM $NaH_2PO_4$ pH 7.4, 500 mM NaCl to remove nonspecifically bound proteins. Bound antibodies were then eluted with fractions of 0.5 ml 0.1 M glycine HCl pH 3 into 1 ml 1 M Tris HCl pH 9, with those containing protein pooled, dialysed into PBS containing 5 mM $NaN_3$, concentrated to about 0.24 mg $ml^{-1}$, flash-frozen in 50-µl aliquots and stored at −80 °C. Specific reactivity of the purified immunoglobulins was confirmed by western blotting against various titin fragments containing the kinase as well as control fragments.

## Super-resolution microscopy

Immunofluorescence labelling was carried out on mouse and rabbit psoas myofibrils as previously described[15] using the affinity-purified TK antibody at 1 µg $ml^{-1}$ and Atto647N-labelled anti-rabbit IgG secondary antibody for visualization. Stimulated emission depletion microscopy was carried out on a STEDYCON (Abberior) attached to a Leica TCS SP5 ll confocal microscope. Images were recorded at a pixel size of 15 nm.

## Reporting summary

Further information on research design is available in the Nature Portfolio Reporting Summary linked to this article.

## Data availability

Cryo-ET structures have been deposited to the Electron Microscopy Data Bank under accession numbers (dataset in brackets) EMD-18200 (thin filament consensus map), EMD-16986 (thin filament with masked out tropomyosin), EMD-16987 (thin filament including tropomyosin), EMD-18147 (thin filament composite map), EMD-16991 (M band from the relaxed thick filament), EMD-16993 (crown P1 from the relaxed thick filament), EMD-16990 (crowns P2–A1 from the relaxed thick filament), EMD-16997 (crowns A1–A5 from the relaxed thick filament), EMD-16996 (crowns A5–A7 from the relaxed thick filament), EMD-16995 (crowns A8–A12 from the relaxed thick filament), EMD-16994 (crowns A11–A15 from the relaxed thick filament), EMD-16992 (crowns A15–A29 from the relaxed thick filament), EMD-18146 (helical reconstruction of the C zone from the relaxed thick filament), EMD-18198 (helical extrapolation of the C zone from the relaxed thick filament). Representative tomograms have been deposited under accession numbers EMD-16989 (tomogram of sarcomere M band to C zone from mouse cardiac muscle) and EMD-16988 (tomogram of sarcomere C zone from mouse cardiac muscle). The atomic coordinates of the protein structures have been submitted to the Protein Data Bank under accession codes (dataset in brackets) 8Q4G (thin filament from the relaxed mouse cardiac muscle), 8Q6T (thick filament helically reconstructed from the C zone of the relaxed mouse cardiac muscle). We used the following previously published structures for modelling and comparisons: 5TBY and 6KN7.

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

**Acknowledgements** We thank D. Prumbaum and O. Hofnagel for assistance with collection of cryogenic electron microscopy data, A. Oelschläger for scripting the calculation of myosin head orientation and R. S. Goody, F. Schnorrer and W. Oosterheert for critical proofreading of the manuscript. This work was supported with funds from the Max Planck Society (to S.R.), the Wellcome Trust (Collaborative Award in Sciences 201543/Z/16/Z to S.R. and M. Gautel), the European Research Council under the European Union's Horizon 2020 Programme (ERC-2019-SyG, grant No. 856118 to S.R. and M. Gautel) and the Medical Research Council (MR/R003106/1 to M. Gautel and A.L.K.). M. Grange was supported by an EMBO Long-Term Fellowship. M. Gautel holds the BHF Chair of Molecular Cardiology. Open access funding was provided by the Max Planck Society.

**Author contributions** S.R. designed and supervised the study. D.T. carried out cryo-FIB milling, cryo-ET experiments and subtomogram averaging, analysed data and prepared figures. Z.W. guided the initial reconstruction of the thin filament. S.T., T.W. and M.S. wrote scripts for particle re-extraction and re-projection. Z.W., S.T. and M. Grange optimized initial FIB-milling and cryo-ET acquisition. A.L.K. and M. Gautel developed methods and prepared myofibril samples. M.R. cloned, expressed, purified and verified the anti-TK antibodies. P.B. carried out super-resolution microscopy experiments. D.T. and S.R. wrote the manuscript with contributions from all authors.

**Funding** Open access funding provided by Max Planck Society.

**Competing interests** The authors declare no competing interests.

**Additional information**
**Correspondence and requests for materials** should be addressed to Stefan Raunser.

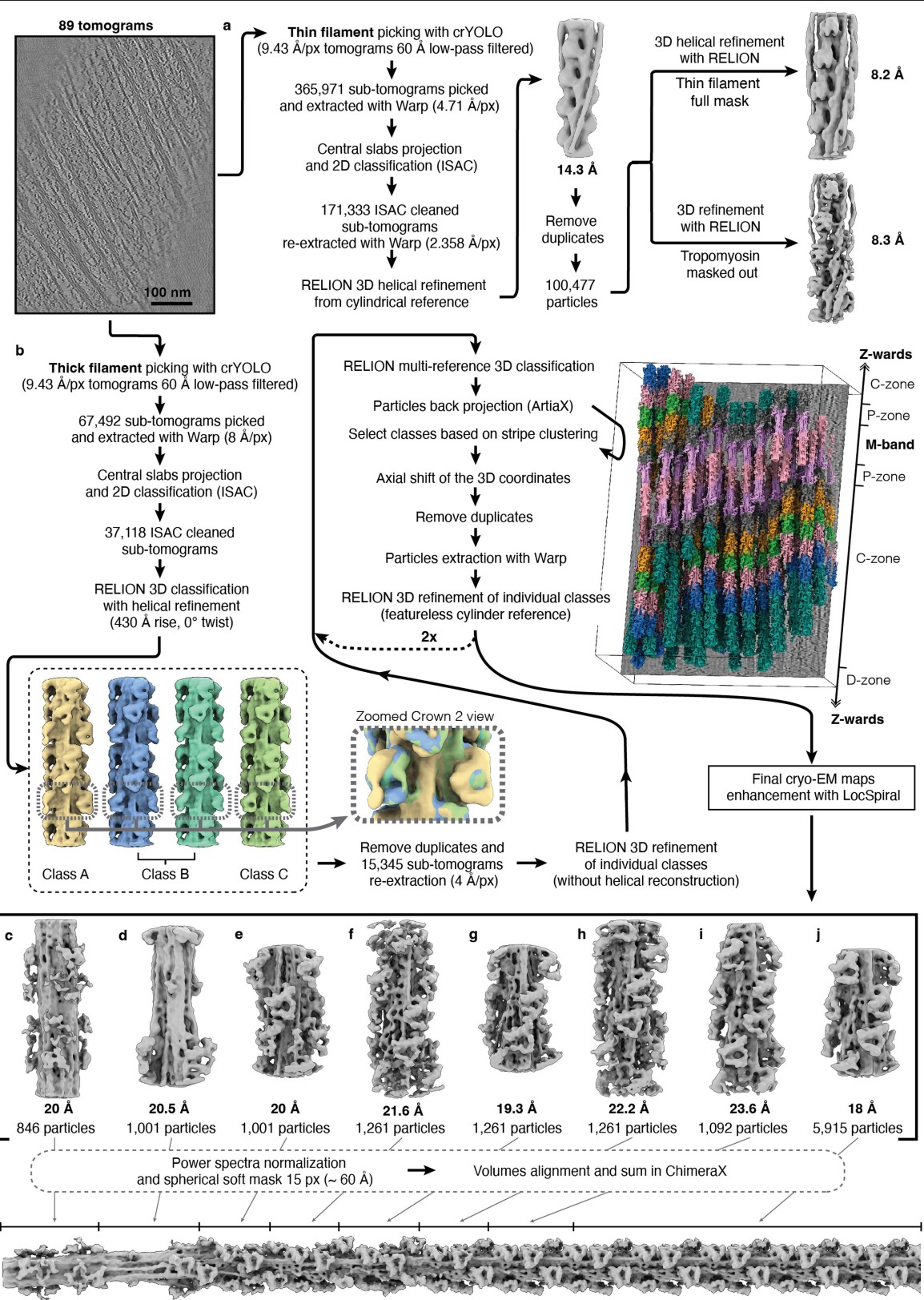

**Extended Data Fig. 1 | Data acquisition and processing pipeline. a,b**, Acquisition and processing of the thin (**a**) and thick filament (**b**). **c-j**, Cryo-EM enhanced maps for: M-band (**c**), A-M transition to crown P1 (**d**), from crown P2 to A1 (**e**), from crown A1 to A5 (**f**), from crown A5 to A7 (**g**), from crown A8 to A12 (**h**), from crown A11 to A15 (**i**), and the combined segments from crown A15 to A29 (**j**).

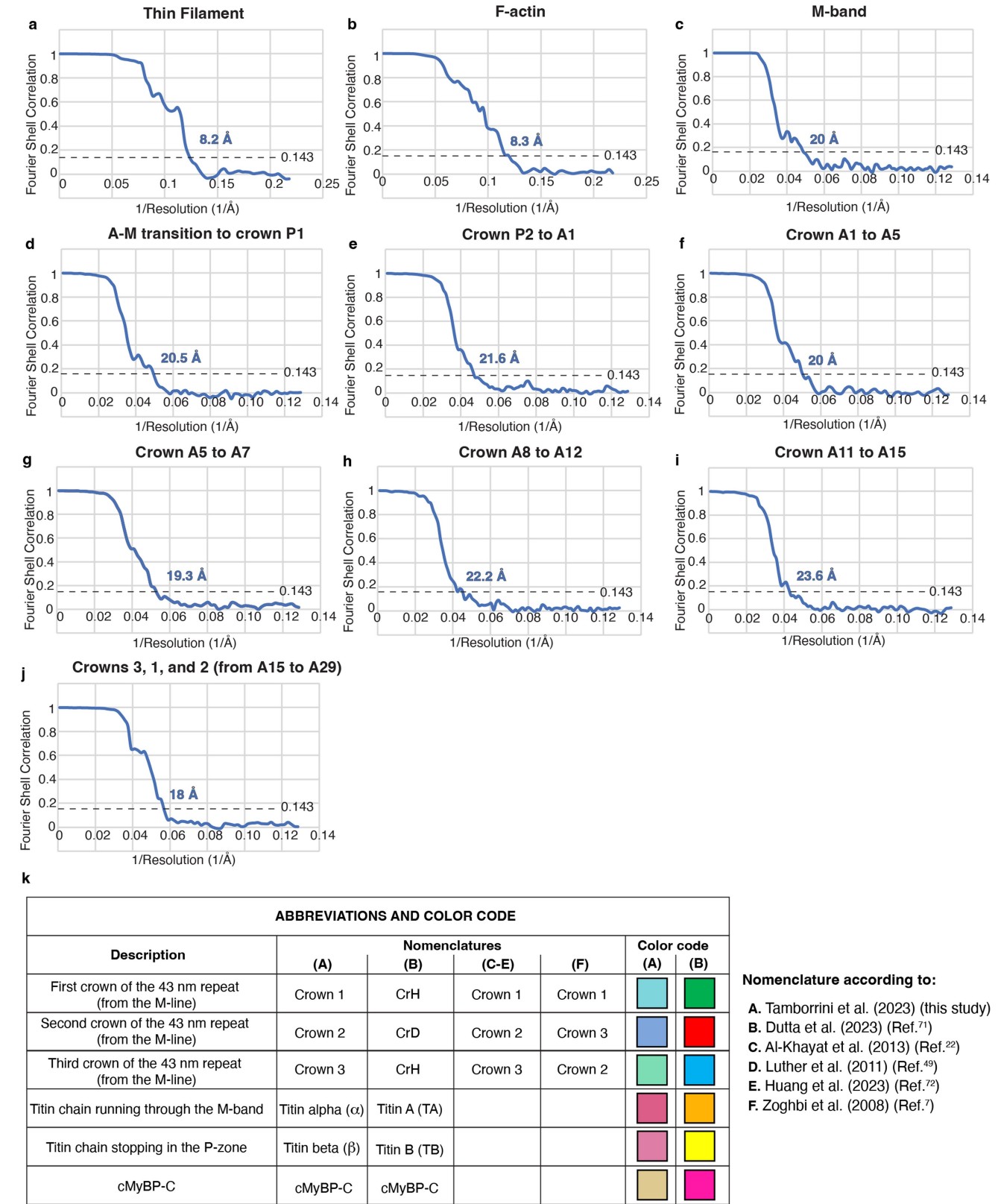

**Extended Data Fig. 2 | Gold-standard FSC curves and nomenclature comparison with other studies. a,b**, Gold-standard FSC curves of cardiac native thin filament (**a**), and F-actin, after masking out tropomyosin (**b**). **c-j**, Gold-standard FSC curves of the different segments of the relaxed thick filament: M-band (**c**), A-M transition to crown P1 (**d**), from crown P2 to A1 (**e**), from crown A1 to A5 (**f**), from crown A5 to A7 (**g**), from crown A8 to A12 (**h**), from crown A11 to A15 (**i**), and the combined segments from crown A15 to A29 (**j**). **k**, Abbreviations and color codes for the structural elements of the thick filament. This table is meant to facilitate the comparisons between nomenclature of this study with other published structural studies. Additionally, we also provide a color code comparison between this study and Dutta et al. (2023)[71]. The full CMYK palette for this study can be found in the Methods.

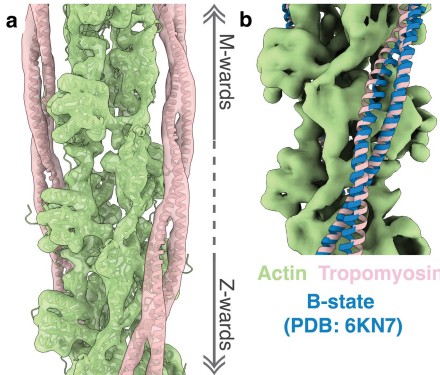

**a**

M-wards

Z-wards

**b**

**Actin** **Tropomyosin**
**B-state**
**(PDB: 6KN7)**

**Extended Data Fig. 3 | In situ structure of the thin filament in the Ca²⁺ free state. a**, Subtomogram-averaged structure of the thin filament in the relaxed sarcomere. **b**, Model of the B-state of tropomyosin (PDB: 6KN7) is shown for comparison.

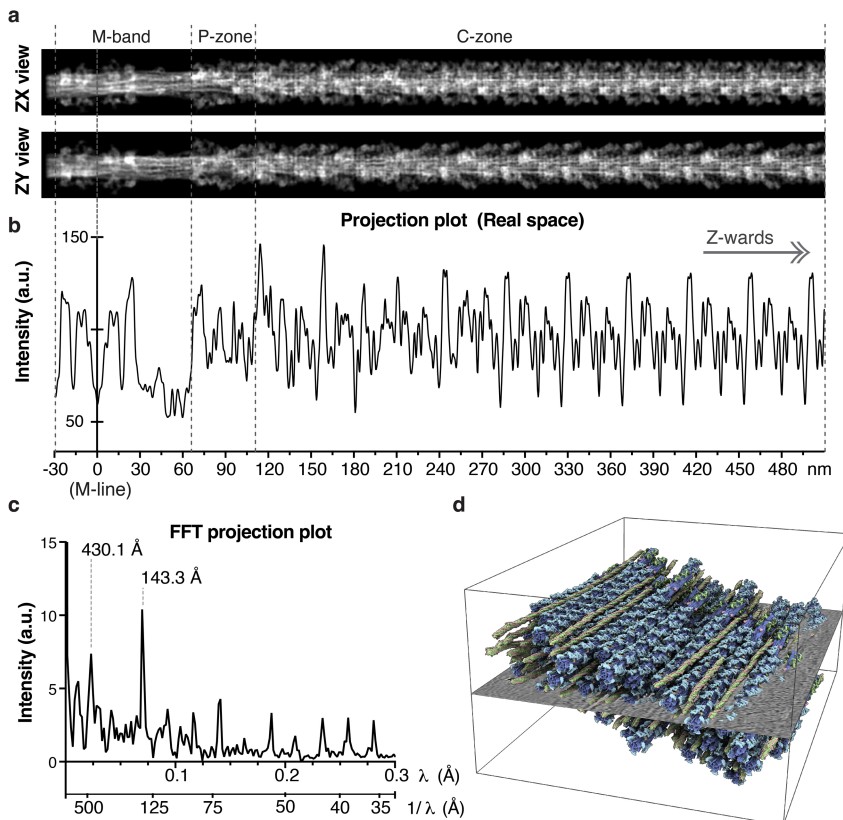

**Extended Data Fig. 4 | Thick filament density projection and profile. a**, The three-dimensional map of the thick filament was projected by summing the intensities of the voxels along the x axis or the y axis. **b**, Intensity profile along the x axis, reveals the periodicity and spacing of the repeating units in the thick filament, in agreement with data from refrozen Tokuyasu cryosections of relaxed rat cardiac muscle[72]. **c**, The Fast Fourier Transform (FFT) of the intensity profile shows the peaks resulting from the repetitive nature of its component. The 430 Å peak is associated to the C-type super-repeats while the 143.3 Å peak is associated to the crowns' layers and was used to calibrate the pixel size of the reconstruction. **d**, Tomographic volume of a cardiac sarcomere A-band showing thick and thin filaments back-plotted into the reconstructed volume with a single tomographic slice at the center for orientation.

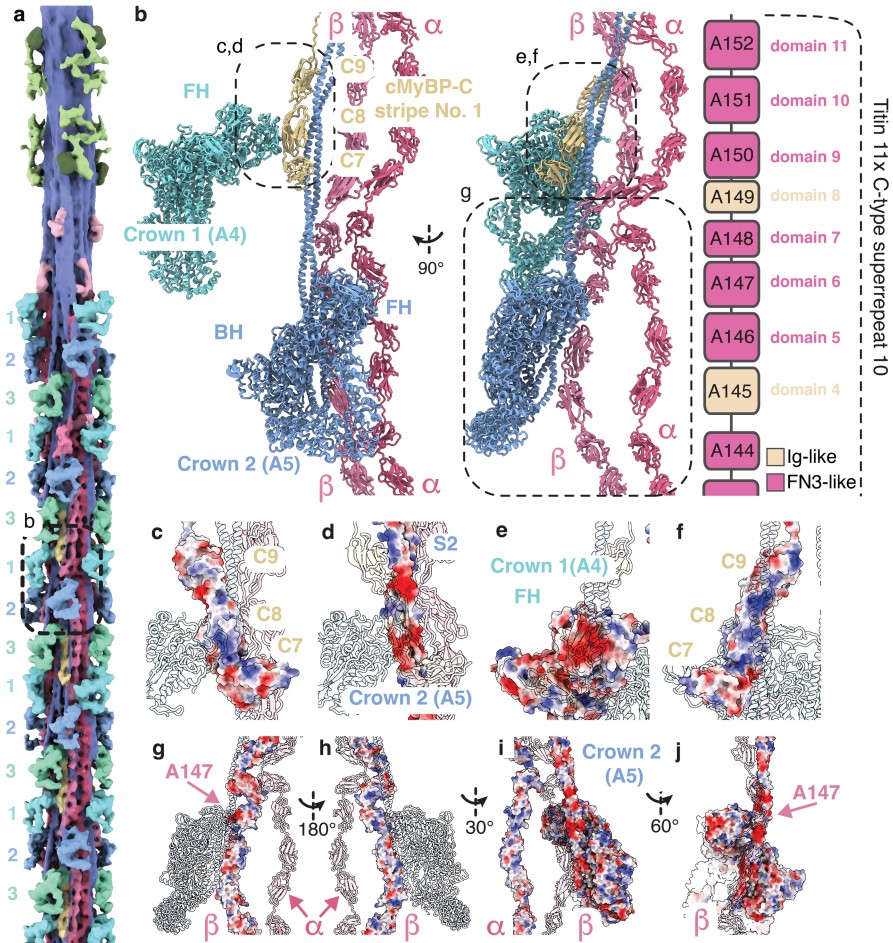

**Extended Data Fig. 5 | The OFF state of crowns 1 and 2 is stabilized by interactions with cMyBP-C and titin. a**, An overview of the thick filament from the M-band to cMyBP-C stripe No. 4 is provided for context. **b**, Simplified molecular model and domain map of the titin C-type super-repeat 10 region showcasing the stabilization of the FHs of crown 1 and crown 2 by the C8 domain of cMyBP-C and the A147 of titin-β, respectively. **c-f**, The FH of crown 1 binds stably to the C8 domain of cMyBP-C via an extensive network of complementary charges (**c,d**), while C8 and C9 bind to the first 80 amino acids of crown 2 tails (**e,f**). **g-j**, The same tail region interacts with titin-β domains (**g,h**). The OFF state of crown 2 is further stabilized by interactions with A147 of titin-β (**i,j**).

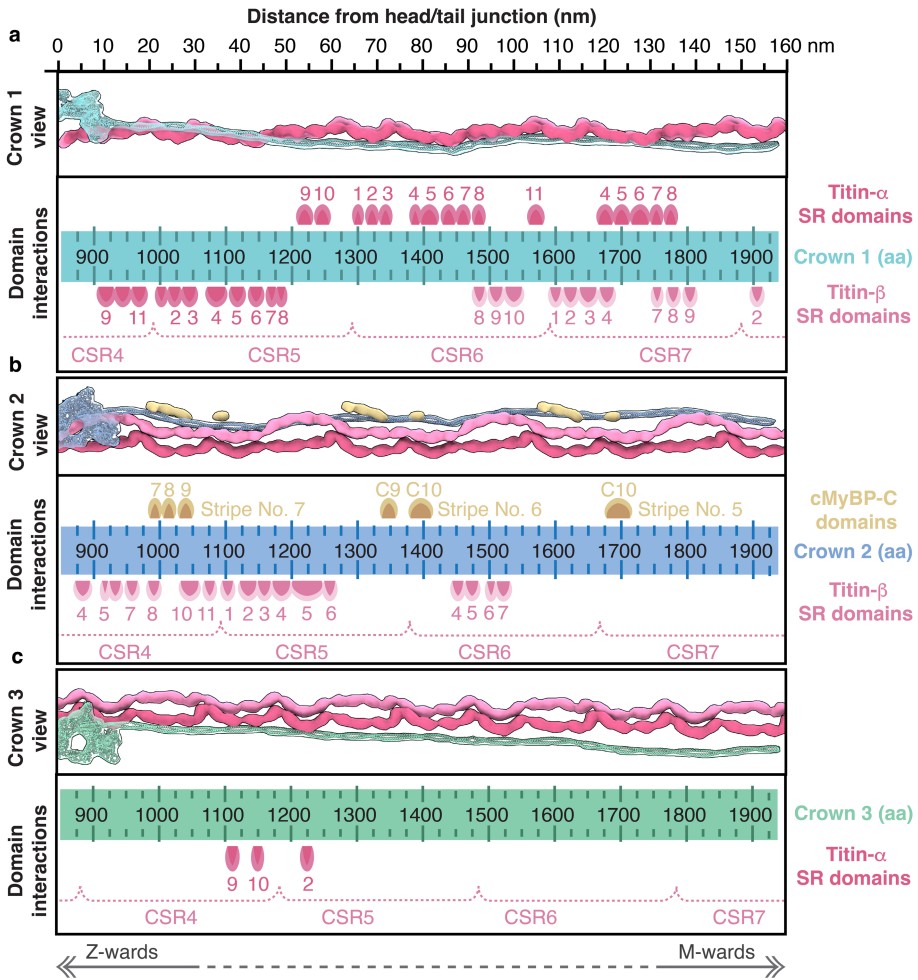

**Extended Data Fig. 6 | Titin domain interactions with myosin tails. a**, The model of crown 1 is shown alongside the densities of titin C-type super-repeats (CSR). To allow a clearer interpretation, a binding site map was generated to depict the amino acid residues of the myosin tail and their distance from the head-tail junction. **b**, The arrangement and interactions of crown 2 are visualized alongside the densities of cMyBP-C and using the same approach as in (**a**). **c**, Similarly, the interaction of crown 3 with titin is shown.

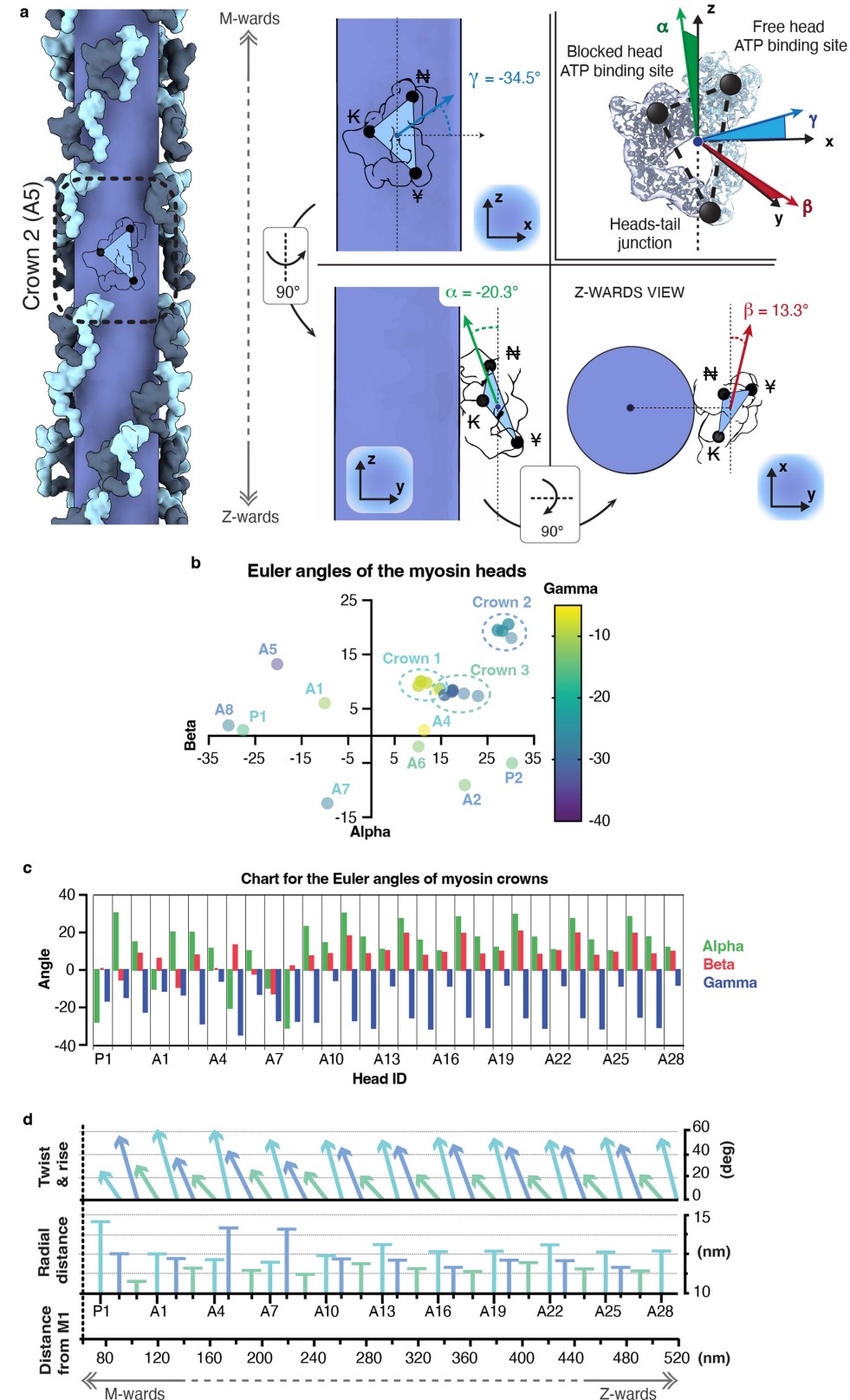

**Extended Data Fig. 7** | See next page for caption.

**Extended Data Fig. 7 | Position and orientation of myosin heads at each crown. a**, The arrangement of myosin heads deviates from a canonical helical symmetry. To quantitatively describe this arrangement, the thick filament was modeled as a cylinder, and the myosin IHMs were represented as triangles (see Methods). The Euler angles of crown 2-A5 are shown as a representative example. **b**, Colored scatter plot of the myosin crowns from P1 to A28. Alpha and beta angles are plotted along the x and y axes, respectively, while the gamma angle is encoded by the color gradient. Plotting of the IHMs generates three orientation clusters for the crowns from A10 to A28, while crowns associated to the P-zone or the first three C-type titin super-repeat show less consistent orientation. **c**, A more detailed representation of (**b**) is shown in a colored histogram. **d**, Using the cylindrical coordinates for centroid of the triangles, we obtained the cylindrical coordinates for each myosin crown and plotted their twist, rise, and radial distance (see Methods). In these stacked charts we plotted each crown according to their distances from the M-band in nanometers (x-axis). The upper chart provides a visual representation of the twist (degrees; y-axis) and rise (nanometers; x-axis) of each crown relative to its Z-ward neighboring crown. The lower chart shows the radial distance of each crown (nanometers; y-axis). Similar to the Euler angle distribution, a more consistent pseudo-helical pattern appears from A10 Z-wards. P1, A5, and A8, show outstanding orientations and radial distances, indicative of a more outward projected configuration.

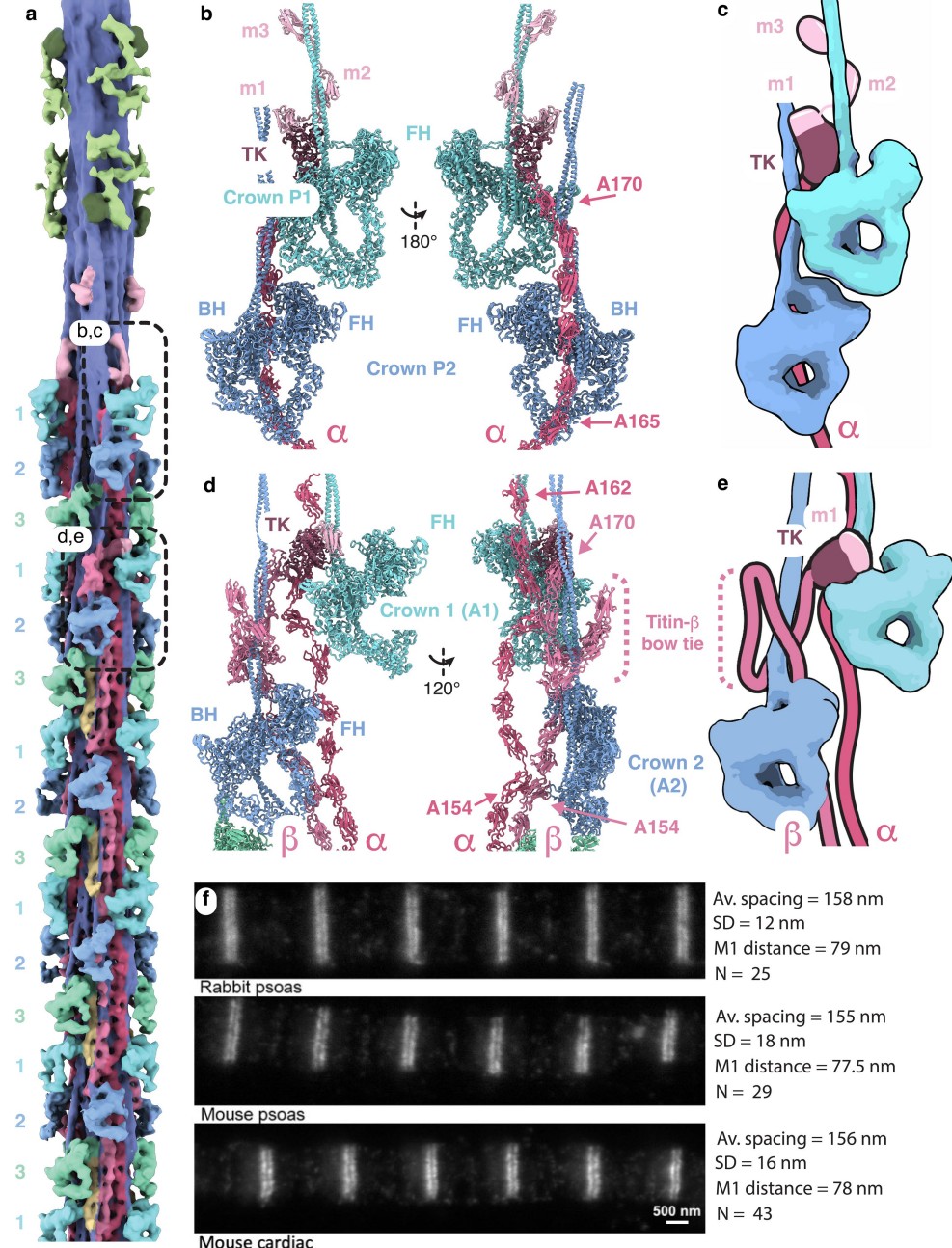

**Extended Data Fig. 8 | The P-zone of the thick filament. a**, An overview of the thick filament from the M-band to cMyBP-C No. 4 is provided for context. **b**, The P-zone of titin-α is located behind the first two crowns, P1 and P2 while the TK domain, m2, and m3 are binding to the head and tail of P1. The m1 domain is interacting with the tail of P2. Notably, there is no density associated to titin-β domains in this region. **c**, Schematic of the organization of titin-α and its relative position to the P1 and P2 crowns. **d**, From titin C-type super-repeat 2 M-wards, the titin-β chain runs alongside the titin-α chain up to the domain A154 of titin. From A162, only the titin-α chain can be traced. Atop the IHM of crown A2, we observed additional globular densities that we assigned to titin-β

forming a flexible bow tie-shaped structure, that projects outwards, and wraps around the tail of crown A2. An additional density atop the motor domain of A1 matches the size and shape of the TK-m1 complex. In this configuration, m1 is positioned on the BH of crown A2. **e**, Schematic of the crowns A1 and A2 region, showing the arrangement of the titin chains and the titin-β chain arrangements around the tail of crown A2. **f**, Super-resolution microscopy localization of the TK domain showing STED imaging of titin kinase in myofibrils from different muscles labeled with affinity-purified anti-TK antibody. For all samples, we report the measurements of doublet spacing, placing TK at 79 nm from the M1 line. SD: standard deviation.

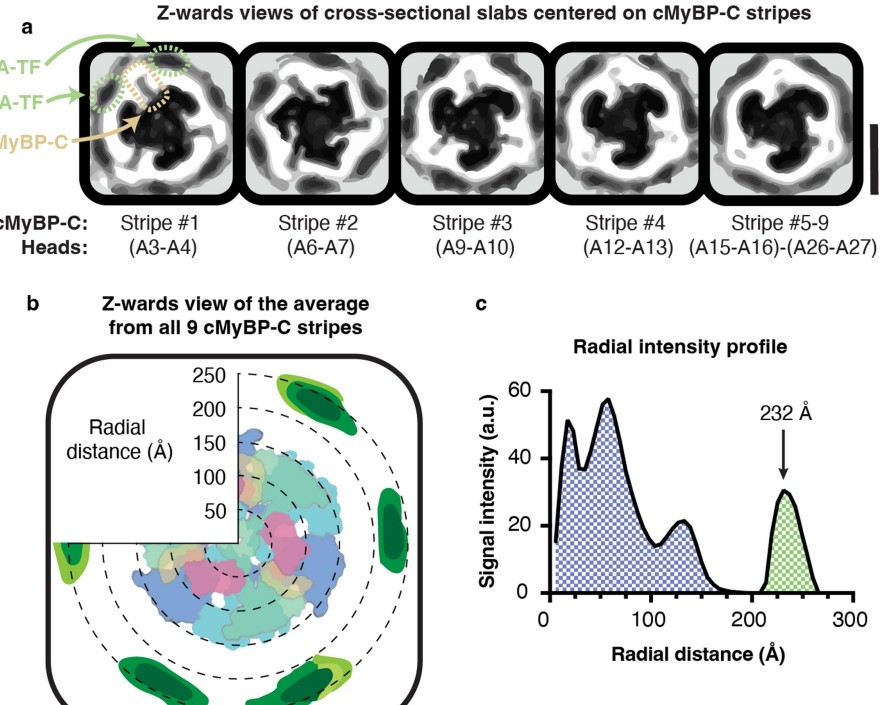

**a** Z-wards views of cross-sectional slabs centered on cMyBP-C stripes

PA-TF
NA-TF
cMyBP-C

| cMyBP-C: | Stripe #1 | Stripe #2 | Stripe #3 | Stripe #4 | Stripe #5-9 |
|---|---|---|---|---|---|
| Heads: | (A3-A4) | (A6-A7) | (A9-A10) | (A12-A13) | (A15-A16)-(A26-A27) |

**b** Z-wards view of the average from all 9 cMyBP-C stripes

Radial distance (Å)

**c** Radial intensity profile

Signal intensity (a.u.)

232 Å

Radial distance (Å)

**Extended Data Fig. 9 | C-zone lattice organization. a**, The bridging domains of cMyBP-C exhibit a high degree of flexibility. To visualize the relative position of the thin filaments and the bridging cMyBP-C region with respect to the thick filament core, we centered each of the nine stripes individually on the cMyBP-C C7 plane and projected the resulting central 40 Å slab along the z-axis. Our analysis showed that stripes Nos. 4 to 9 projected into a region equidistant from both the negative azimuthal angle thin filament (NA-TF) and the positive azimuthal angle thin filament (PA-TF), indicating that cMyBP-C may bind to either of them with equal likelihood. In contrast, stripes No. 2 showed a preferential orientation towards NA-TF, while stripe No. 1 exhibited a clear preferential bias towards PA-TF. **b**,**c**, We generated a single contour map by averaging all nine stripes together, which was then used to obtain a model for the determination of the average thin filament distance and azimuthal angles for the entire 430 Å repeat. Our analysis revealed that the center-to-center distance from thick to thin filament was 232 Å, as demonstrated by the radial profile of the signal intensity of a full 430 Å repeat. Scale bars: 25 nm.

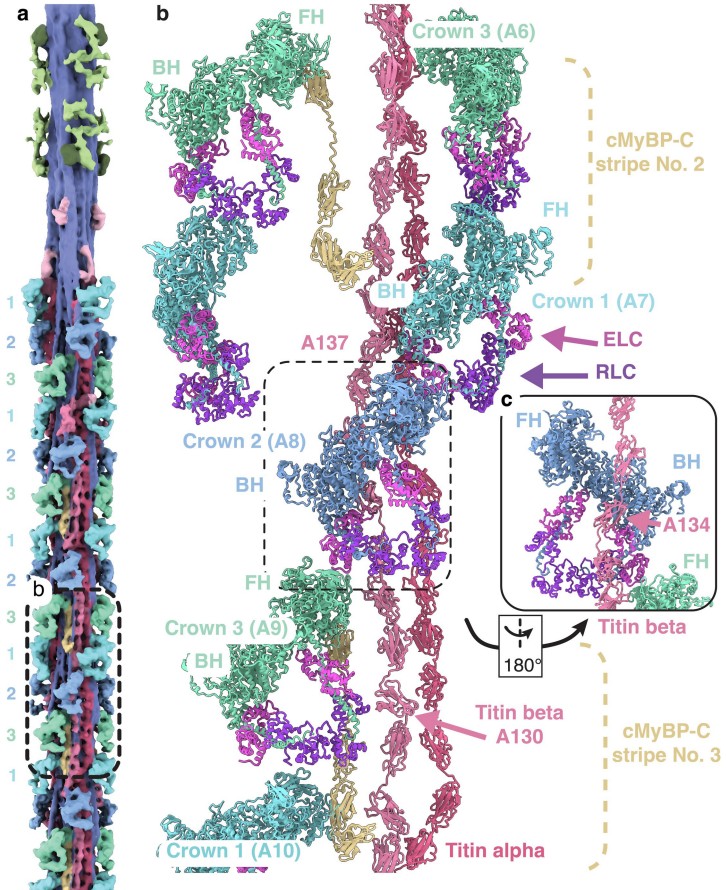

**Extended Data Fig. 10 | Myosin OFF state is stabilized by myosin light chains at titin C-type super-repeat 9. a**, An overview of the thick filament from the M-band to cMyBP-C stripe No. 4 is provided for context. **b**, Simplified overview of a single asymmetrical unit at CSR9, showing the molecular arrangement of the myosin crowns, titin and cMyBP-C. In contrast to all other cMyBP-C, the crown 1 does not interact with cMyBP-C CTD at stripe No. 2, instead its FH binds to the RLC from crown 3 and its BH interacts with titin A137. Crown 2 orientation is also atypical as its FH interacts with the disordered region of the ELC of crown 1 and its BH binds to titin A134 (**c**). Similarly, the FH of crown 3 (A9) binds to the ELC of crown 2 and it is also stabilized by an interaction of its RLC with titin-β A130.

# Reporting Summary

## Statistics

For all statistical analyses, confirm that the following items are present in the figure legend, table legend, main text, or Methods section.

| n/a | Confirmed | |
|---|---|---|
| ☐ | ☒ | The exact sample size (*n*) for each experimental group/condition, given as a discrete number and unit of measurement |
| ☒ | ☐ | A statement on whether measurements were taken from distinct samples or whether the same sample was measured repeatedly |
| ☒ | ☐ | The statistical test(s) used AND whether they are one- or two-sided *Only common tests should be described solely by name; describe more complex techniques in the Methods section.* |
| ☒ | ☐ | A description of all covariates tested |
| ☒ | ☐ | A description of any assumptions or corrections, such as tests of normality and adjustment for multiple comparisons |
| ☐ | ☒ | A full description of the statistical parameters including central tendency (e.g. means) or other basic estimates (e.g. regression coefficient) AND variation (e.g. standard deviation) or associated estimates of uncertainty (e.g. confidence intervals) |
| ☒ | ☐ | For null hypothesis testing, the test statistic (e.g. *F*, *t*, *r*) with confidence intervals, effect sizes, degrees of freedom and *P* value noted *Give P values as exact values whenever suitable.* |
| ☒ | ☐ | For Bayesian analysis, information on the choice of priors and Markov chain Monte Carlo settings |
| ☒ | ☐ | For hierarchical and complex designs, identification of the appropriate level for tests and full reporting of outcomes |
| ☒ | ☐ | Estimates of effect sizes (e.g. Cohen's *d*, Pearson's *r*), indicating how they were calculated |

*Our web collection on statistics for biologists contains articles on many of the points above.*

## Software and code

Policy information about availability of computer code

| Data collection | Cryo-electron tomograms were collected using Serial-EM version 3.8 |
|---|---|
| Data analysis | CTF fitting and motion correction performed in Warp v1.0.9. Tilt series alignment performed in IMOD 3dmod v4.10.51. Tomogram reconstruction and sub-tomogram extraction performed in Warp v1.0.9. Particles picking performed with crYOLO v1.8. Sub-tomogram averaging and post-processing executed in Relion v3.1.0. Map enhancement perfomed with LocSpiral. Tomogram denoising performed with cryoCARE. Protein structure prediction performed with AlphaFold V2.1.1. Segments of the same protein chains were linked together in COOT v0.8.9.2. Flexible fitting molecular dynamic performed with NAMDinator. Figures and videos that depict cryo-EM density maps and protein structures prepared using UCSF ChimeraX v1.5 and ArtiaX extention. Data plotting was done in GrapPad Prism v9 (GraphPad Software). |

For manuscripts utilizing custom algorithms or software that are central to the research but not yet described in published literature, software must be made available to editors and reviewers. We strongly encourage code deposition in a community repository (e.g. GitHub). See the Nature Portfolio guidelines for submitting code & software for further information.

## Data

Policy information about availability of data

 All manuscripts must include a data availability statement. This statement should provide the following information, where applicable:
- Accession codes, unique identifiers, or web links for publicly available datasets
- A description of any restrictions on data availability
- For clinical datasets or third party data, please ensure that the statement adheres to our policy

Cryo-ET structures have been deposited to the Electron Microscopy Data Bank (EMDB) under accession numbers (dataset in brackets): EMD-18200 (thin filament consensus map) [https://www.ebi.ac.uk/emdb/EMD-18200], EMD-16986 (thin filament with masked out tropomyosin) [https://www.ebi.ac.uk/emdb/EMD-16986], EMD-16987 (thin filament including tropomyosin) [https://www.ebi.ac.uk/emdb/EMD-16987], EMD-18147 (thin filament composite map) [https://www.ebi.ac.uk/emdb/EMD-18147], EMD-16991 (M-band from the relaxed thick filament) [https://www.ebi.ac.uk/emdb/EMD-16991], EMD-16993 (Crown P1 from the relaxed thick filament) [https://www.ebi.ac.uk/emdb/EMD-16993], EMD-16990 (Crowns P2-A1 from the relaxed thick filament) [https://www.ebi.ac.uk/emdb/EMD-16990], EMD-16997 (Crowns A1-A5 from the relaxed thick filament) [https://www.ebi.ac.uk/emdb/EMD-16997], EMD-16996 (Crowns A5-A7 from the relaxed thick filament) [https://www.ebi.ac.uk/emdb/EMD-16996], EMD-16995 (Crowns A8-A12 from the relaxed thick filament) [https://www.ebi.ac.uk/emdb/EMD-16995], EMD-16994 (Crowns A11-A15 from the relaxed thick filament) [https://www.ebi.ac.uk/emdb/EMD-16994], EMD-16992 (Crowns A15-A29 from the relaxed thick filament) [https://www.ebi.ac.uk/emdb/EMD-16992], EMD-18146 (Helical reconstruction of the C-zone from the relaxed thick filament) [https://www.ebi.ac.uk/emdb/EMD-18146], EMD-18198 (Helical extrapolation of the C-zone from the relaxed thick filament) [https://www.ebi.ac.uk/emdb/EMD-18198]. Representative tomograms have been deposited under accession numbers EMD-16989 (Tomogram of sarcomere M-band to C-zone from mouse cardiac muscle) [https://www.ebi.ac.uk/emdb/EMD-16989] and EMD-16988 (Tomogram of sarcomere C-zone from mouse cardiac muscle) [https://www.ebi.ac.uk/emdb/EMD-16988].

The atomic coordinates of the protein structures have been submitted to the Protein Data Bank under accession codes (dataset in brackets): 8Q4G (Thin filament from the relaxed mouse cardiac muscle) [https://doi.org/10.2210/pdb8Q4G/pdb], 8Q6T (Thick filament helically reconstructed from the C-zone of the relaxed mouse cardiac muscle) [https://doi.org/10.2210/pdb8Q6T/pdb]. We used the following previously published structures for modelling and comparisons: 5TBY and 6KN7. Source data are provided with this paper.

## Human research participants

Policy information about studies involving human research participants and Sex and Gender in Research.

| Reporting on sex and gender | *Use the terms sex (biological attribute) and gender (shaped by social and cultural circumstances) carefully in order to avoid confusing both terms. Indicate if findings apply to only one sex or gender; describe whether sex and gender were considered in study design whether sex and/or gender was determined based on self-reporting or assigned and methods used. Provide in the source data disaggregated sex and gender data where this information has been collected, and consent has been obtained for sharing of individual-level data; provide overall numbers in this Reporting Summary. Please state if this information has not been collected. Report sex- and gender-based analyses where performed, justify reasons for lack of sex- and gender-based analysis.* |
|---|---|
| Population characteristics | *Describe the covariate-relevant population characteristics of the human research participants (e.g. age, genotypic information, past and current diagnosis and treatment categories). If you filled out the behavioural & social sciences study design questions and have nothing to add here, write "See above."* |
| Recruitment | *Describe how participants were recruited. Outline any potential self-selection bias or other biases that may be present and how these are likely to impact results.* |
| Ethics oversight | *Identify the organization(s) that approved the study protocol.* |

Note that full information on the approval of the study protocol must also be provided in the manuscript.

# Field-specific reporting

Please select the one below that is the best fit for your research. If you are not sure, read the appropriate sections before making your selection.

☒ Life sciences  ☐ Behavioural & social sciences  ☐ Ecological, evolutionary & environmental sciences

For a reference copy of the document with all sections, see nature.com/documents/nr-reporting-summary-flat.pdf

# Life sciences study design

All studies must disclose on these points even when the disclosure is negative.

| Sample size | We collected 89 tomographic volumes. Sample sizes for the ten cryo-EM map obtained in this study: structure of F-actin without tropomyosin resulted from the averaging of 100,447 particles, structure of F-actin with tropomyosin resulted from the averaging of 100,447 particles, |
|---|---|

structure of the thick filament M-band resulted from the averaging of 846 particles, structure of the thick filament Crown P1 resulted from the averaging of 1001 particles, structure of the thick filament Crowns P2-A1 resulted from the averaging of 1001 particles, structure of the thick filament Crowns A1-A5 resulted from the averaging of 1261 particles, structure of the thick filament Crowns A5-A7 resulted from the averaging of 1261 particles, structure of the thick filament Crowns A8-A12 resulted from the averaging of 1261 particles, structure of the thick filament Crowns A11-A15 resulted from the averaging of 1092 particles, structure of the thick filament Crowns A15-A29 resulted from the averaging of 5915 particles.

| Data exclusions | During the cryo-EM image processing, particles that represented false picks were discarded through 2D and 3D classification procedures. This process, which is required to obtain high-resolution reconstructions, is a standard procedure in cryo-EM image processing. Duplicated particles were removed. |
| Replication | Cryo-EM grids of vitrified material were obtained in a single plunging session. It is unattainable from a time and cost perspective to repeat cryo-EM data collection and processing on the exact same sample. |
| Randomization | For the 3D refinement of cryo-EM structures, particles were randomly split into two half sets. For all other experiments, randomization was not required because all data were used in the analysis. Covariates were not controlled. |
| Blinding | This study does not involve any experiments where blinding would be applicable. |

# Reporting for specific materials, systems and methods

We require information from authors about some types of materials, experimental systems and methods used in many studies. Here, indicate whether each material, system or method listed is relevant to your study. If you are not sure if a list item applies to your research, read the appropriate section before selecting a response.

## Materials & experimental systems

| n/a | Involved in the study |
|---|---|
| ☐ | ☒ Antibodies |
| ☒ | ☐ Eukaryotic cell lines |
| ☒ | ☐ Palaeontology and archaeology |
| ☐ | ☒ Animals and other organisms |
| ☒ | ☐ Clinical data |
| ☒ | ☐ Dual use research of concern |

## Methods

| n/a | Involved in the study |
|---|---|
| ☒ | ☐ ChIP-seq |
| ☒ | ☐ Flow cytometry |
| ☒ | ☐ MRI-based neuroimaging |

## Antibodies

| Antibodies used | Affinity-purified rabbit polyclonal antibody against titin kinase |
| Validation | Antibody was affinity-purified against recombinant rat titin kinase and validated by Western blot against recombinant titin fragments containing the kinase domain as well as negative controls (fragments not containing the kinase outside of the P-zone/M-band transition). |

## Animals and other research organisms

Policy information about studies involving animals; ARRIVE guidelines recommended for reporting animal research, and Sex and Gender in Research

| Laboratory animals | BALB/c mice were used for myofibril preparations. |
| Wild animals | *Provide details on animals observed in or captured in the field; report species and age where possible. Describe how animals were caught and transported and what happened to captive animals after the study (if killed, explain why and describe method; if released, say where and when) OR state that the study did not involve wild animals.* |
| Reporting on sex | Sex was not considered as there is no evidence that myofibril structure differs between male and female. |
| Field-collected samples | *For laboratory work with field-collected samples, describe all relevant parameters such as housing, maintenance, temperature, photoperiod and end-of-experiment protocol OR state that the study did not involve samples collected from the field.* |
| Ethics oversight | Animals were sacrificed in a schedule-1 procedure by cervical dislocation following licensed procedures approved by King's College London ethics committee and the Home Office UK. |

Note that full information on the approval of the study protocol must also be provided in the manuscript.

