## [Peer Review File · Nature]

Manuscript Title: Structure of the native myosin filament in the relaxed cardiac sarcomere

Reviewer Comments & Author Rebuttals

Reviewer Reports on the Initial Version:

Referees' comments:

Referee #1 (Remarks to the Author):

A. Summary of the key results

The manuscript describes a FIB SEM / cryo-electron tomography study of mouse cardiac muscle in the relaxed state. The resulting structural models, particularly of the myosin thick filament, but also its interaction with surrounding thin filaments are stunning and one of the most exciting pieces of structural biology I have seen to date. The combination of sub-tomogram averaging and interpretation of raw tomograms provides an unprecedented insight into the whole structure.

Both this paper, and the co-submitted manuscript by the Craig lab, explain for the first time how different “crowns” of myosin heads have the potential to behave differently in the muscle. They also provide an elegant description of how the giant titin proteins run through the thick filament. This manuscript, by imaging the entire thick filament in its native context also provides fascinating information about the thick/thin filament interactions mediated by the MyBP-C protein and the structure of the M and P zones of the thick filament.

B. Originality and significance: if not novel, please include reference

The work is exceptionally novel and significant. It answers years-worth of questions regarding the organisation of myosin's in muscle and provides the basis of many studies in the future.

C. Conclusions: robustness, validity, reliability

The resolution of the structure is by necessity quite low in places, but the authors are extremely careful in their interpretations and are guided by both previous structures and alpha-fold. The only place where I felt their interpretation was less clear was the model they build of the titin filament. I miss, given the current resolution, how they are able to assign the Ig-like and Fn3-like domains of the titin superrepeat to the density (e.g. page 8 line 221 – needs more explanation and a figure showing density – in addition to the movie).

D. Suggested improvements: experiments, data for possible revision

Besides the above, no new experiments or analysis are required. However as described below I would recommend the manuscript and figures would benefit from some re-organisation.

E. Clarity and context: lucidity of abstract/summary, appropriateness of abstract, introduction and conclusions

- 1) Mostly the manuscript is well written and appealing to a general audience.
- 2) There are a few minor grammatical errors: e.g. in the abstract “thick filaments” and “its” rather than “their”.
- 3) The one big problem with the manuscript is that the Figure and Extended Data Figure references are jumbled up and do not appear to correspond to what is being described in the text. E.g. Page 7 line 174 (Fig 5a does not show what is described), Page 8 & 9: Titin section: All figure references seem to be wrong/not show what is reported in the text. Page 12 & 13. Same problem.
- 4) The flow of the manuscript needs to be reworked slightly to follow the Figures. Currently it jumps over Fig 4a-d, straight to Fig4e.
- 5) I would bring some parts of the Titin description, currently in the Supplement, into the main text. It was very hard to follow the description without these figures.
- 6) Some specific figure comments: I really like the color scheme in general. It is calm and elegant. In Fig2f it is hard to tell the difference between Titin alpha and beta (label or change color slightly?). In Fig2i the numbering 1st, 2nd etc doesn't match the text (1,2 etc). Perhaps put these cMyPB-C stripes to the left of “g” where it will be clearer what they refer to? Fig 3c: Label “Blocked Head (BH)” etc to make it clearer what “BH” corresponds to. Fig4: perhaps move Fig4d and 4h near to each other. Also flip 4h so it is in the same orientation? Fig4f – what is the yellow density contacting Crown1 and Crown3? Label? Fig6: label what the green density in Fig6c is – also refer to it in the figure legend?
- 7) Page 5, line 120: the title says the order is myosin, cMyBP-C, titin. In contrast the text goes myosin, titin, cMyBP-C.
- 8) Similarly the order of the Figures is roughly: Overview, myosin heads(3), cMyBP-C (4), myosin tails (5), M-band (6). With no Titin figure. The order of the text sections is Overview: myosin tails, myosin heads, titin, M-band, cMyBP-C. I understand it is a complicated system with lots of overlap between what needs to be talked about, but I think the manuscript will be easier to read if it follows a more fixed order and that order was followed throughout.
- 9) Two very minor points:
 - a. Could the authors explain explicitly that the “interacting heads motif” is actually a structural conformation of the myosin molecule (rather than a peptide motif).
 - b. I understand the differences in nomenclature between this paper and the Craig paper (Titin L/R vis alpha/beta), Crown 1-3 vis CrH,CrT,CrD etc. It is ok to leave it, but I wonder if there is any chance the authors could confer and come up with a unified nomenclature?

Referee #2 (Remarks to the Author):

This paper presents the discovery of the three dimensional structure of a major part of the mouse cardiac thick filament in the relaxed state. The structures of the myosin, titin and cMyBP-C molecules can all be seen. The 3D arrangement of all three molecules that is revealed include unpredicted features that mean that this work is not merely an extension of previous studies of invertebrate thick filaments but represent a completely new and more complex arrangement than hitherto suspected. All previous models of vertebrate thick filament structure are proven wrong by these new results. A structure for the blocked state thin filament in situ is also a new result of this study. As the work is done on mammalian cardiac myofibrils, the results also have implications for the understanding and treatment of many cardiac diseases, so the results will be of interest to an unusually wide audience.

The results are obtained by an established and reliable workflow, the text is lucid and the Figures effectively communicate the complex results.

The benefits of using cryo-ET on intact myofibrils compared to working with isolated thick filaments are evident from the ability to characterise the interactions of cMyBP-C with both thick and thin filaments as well as by the discovery of perturbations in the structure along the length of the filament that would easily have been missed by averaging the repeating 43nm units. However this does limit the resolution obtained to about 2 nm, which does not reveal secondary structure within the proteins, only their overall shape, and the coiled coil tail of myosin is a smooth cylinder so that the local azimuth of its coiled coil structure cannot be seen. To interpret their results in terms of an atomic model of the structure, the authors make sensible decisions in modelling in known structures and AlphaFold predictions, but it must be recognised that, for instance the orientation of titin and cMyBP-C domains within the blobs of density are best guesses, and are not determined by the EM data itself. For the benefit of the general readership of Nature, the authors should include some such statement at the start of the Results section.

Video 1 shows features of the myofibril specimen that raise some concerns. It shows that the thick filaments are not in transverse register: the bare zones are axially staggered by up to ~300nm in the sample shown, whereas they should be aligned. This indicates that the filaments actively sheared past one another during specimen preparation, probably at the step at which ATP was added to the rigor myofibril preparation, as this induces a brief contraction. A consequence is that the M-line bridges will have been broken and that suggests that the M-line structure reported is not the native one. Likewise, there are thin filaments throughout the bare zone, indicating that the sarcomere length is short, and there is little indication that the thin filaments are located at the trigonal points of the thick filament array as they should be. In consequence thin filaments of the opposite polarity may be present alongside the thick filaments and some of the cMyBP-C bridges described may involve such filaments and are thus likely to be non-physiological. To address these concerns, the authors should report the sarcomere length of the specimens they have imaged, and make counts of the thin to thick filament ratio in their tomograms – is it 2:1 as in vivo, or 4:1 as in over-shortened myofibrils?

These concerns do not however undermine the great advances that are presented on thick filament structure.

Minor points:

There is some confusion in labelling of the two titin chains. In the text (line 230) it is stated that titin alpha runs across the bare zone, but in Fig. 2(f) the label Titin Alpha is dark pink, corresponding to the colour of the titin chain that ends near crown A1. Also in panel (g), the truncated chain appears to be the paler (left-hand) one, i.e. the opposite colouring of panel (f). The authors need to correct this confusion, but also to check every Figure panel to be sure that the colouring of both the protein and the labelling are consistent throughout the paper. Given this mix-up of colouring, they should also check every text statement to make sure that statements about alpha and beta titins do indeed refer to the correct molecule, rather than a statement linked to an incorrectly-coloured molecule. It would also help the reader if the difference in shade of pink of the two titins and all the titin labels was made more pronounced in all the Figures.

Related to the last point, the use of depth-dependent colouring makes a nonsense of the distinction between the two titin chains, because the dark pink titin chain becomes pale pink when it is further away. This renders ED Fig. 5b and, especially, ED Fig 9 almost impossible to decipher. Please turn off depth cueing and replace any panels in which it has been used.

The labelling in the otherwise excellent Fig. 2 should be revised to reduce confusion. In panel (d) the space between successive crown numbers is variable, which I initially thought reflected an actual variation in crown spacing in panel (c), but the correlation is not good. It would be helpful if the numbers were indeed carefully placed so that they did emphasize variable crown spacing, for instance by placing them such that the lower edge of each number was at exactly the same level as the lever-lever junction of each IHM crown. The titin labels (panel (h)) should be changed from C2-C11 to CSR2-CSR11, like in ED Fig7, to reduce confusion with C-stripe numbers. It would also help to change them from prominent black to pink, which will help the reader to notice the relatively visually weak ochre-coloured C-stripe labels to the right (panel (i)). Finally, panels (g-i) are at a slightly larger scale than the rest of the Figure, such that they are in register with panel (f) at the top of the Figure, but are out of sync at the bottom. Please adjust.

Video 2 fails to colour the two titin chains differently in the EM density surface, to match Fig. 2. In correcting this, it won't matter if one of them is then the same colour as the tropomyosin strands, as there's no chance of confusing the two structures.

Line 230: Apart from a few blobs which might be titin or some other myofibrillar protein, I don't see any evidence of titin in the bare zone in Fig 2 (a, b), let alone that it crosses all the way to the opposite P-zone, so what is the basis for the assertion that the alpha chains do this? Is this a conclusion from some other (unreferenced) study by antibody labelling? On a more general point, is the truncation of beta titin that they observe a consequence of the shearing between thick filaments that has disrupted M-line structure? It is possible that beta titin was attached to M-bridges and therefore was snapped apart by shear, or was more susceptible to postmortem proteolysis in the bare zone than alpha titin molecules, with the result that the clump of titin domains near A1 is an

artefactual recoil rather than a native structure.

Location of the titin kinase (TK) domains: By using antibody labelling the site of TK is measured to be 79 nm from the centre of the bare zone. The authors state that this is the position of the TK of the alpha chain, and that the TK of the beta chain is unlabelled. But the putative densities in their EM density map are only about 43nm apart. It is doubtful that their STED imaging is capable of resolving two positions that are so close, since there is only a narrow separation between the two stripes separated by 156 nm across the bare zone. Therefore, how can they rule out that the antibody has labelled both TK domains and the separation measured by STED is the midway point between the alpha and beta copies of TK across the bare zone? If the truncation of beta titin is an artefact (see comment above), this STED result might reflect where the beta titin TK was before M-line disruption.

Lines around 360: Because the thin filament repeats at ~38nm intervals and the cMyBP-C at 43nm intervals, is this flexibility due to the cMyBP-C always seeking out the same target on blocked state thin filaments? - i.e. do the attachment sites of cMyBP-C on thin filaments show a 38nm periodicity?

Line 52: Myosin molecule mass should include the masses of its four light chains, not just its two heavy chains, and therefore be around 530 kDa not 446 kDa.

Line 118: the claim of describing '...the thick filament across the entire sarcomere...' is unjustified as the D-zone is sadly not included in the current study.

Fig. 3 legend needs to state the direction of view of the cross sections – M-wards or Z-wards?

Line 225: Figure referenced should be Fig 2b not 5b.

The legend to ED Fig. 9 (d-f) is too brief – it doesn't say what each of the three panels depicts.

Lines 266 and 269: Here there is reference to a bow-tie feature, supposedly depicted in ED Fig 9d. I think this should be a reference to panel (c) of the Figure in which there is a label 'Titin beta tie', but even there what I see to the left of the label is a U-shaped sequence of domains, not a bow-tie shape. Maybe 'cluster' would be simpler.

Lines 329 and 856: 'interphase' should be 'interface'.

Peter Knight

Referee #3 (Remarks to the Author):

Tamborrini and colleagues use advanced cryo-electron tomography methods to report the organization of the muscle thick filament in its relaxed state in situ. This allows them to provide a holistic description of the components forming the thick filament in their native context, providing yet unknown insights into their interaction during muscle contraction. Together, with other recent studies from the same lab (Wang et al., 2021 & Wang et al., 2022) the authors continue to fill in a complete high-resolution structural picture of the sarcomere architecture. The wealth of biological observations obtained make this a very complex and fascinating study. I applaud the authors in preparing figures that do a good job in guiding the reader through the different aspects of the complex thick filament structure.

The methodological implementation of their established cryo-focused ion beam milling and cryo-ET pipeline is again state-of-the-art and meticulously performed. My review will focus on technological aspects which deserve attention for minor clarifications.

Overall, this is an outstanding study, considering both the biological findings and methodological implementation.

- Thick filament processing:

In Extended Data figure 1 the mentioned differences in crowns of classes A, B and C are difficult to see (projected, retracted IHMs in class A and C, and the fuzzy appearance for crown 2 IHM in class B). Please annotate these differences.

It is also not entirely clear how these different classes are reflected in the different obtained structures. Specifically, class A apparently showed a unique distribution in the sarcomere. I assume this means class B and C did not? Does this reflect something of biological relevance or has this been caused by any limitation in the classification process (e.g. lower SNR of such particles)?

In particular, given another recent study on the cardiac myosin filament (Dutta et al.), where one of the crowns appears to be disordered, one could speculate about whether this structure corresponds to one of the observed classes in the complete in situ thick filament.

It is not entirely clear what the authors refer to with "Through an iterative process of 3D refinement, multi-reference 3D, classification, particles back-projection, and axial shift calculations" (Methods, page 22/line 635). Can the authors explain better what the axial shift rotation calculations are and why they were necessary?

Also, I assume the processing has been done via gold-standard dataset splitting. Could the authors elaborate at which step in the processing the datasets have been separated. This is not an crucial comment, but still would be interesting to know, given the shape of some FSC curves, that show a jump close to the 0.143 criterion (e.g ED Figure 2c,d,i).

The way how the authors have segmented the cMyBP-C links is very creative and clever. However, it would be interesting to know what kind of validation was used to verify that a link between thick and thin filaments has been identified accurately, i.e. was after segmentation every remaining non-assigned density in the tomograms between thick and thin filament automatically assigned to be a

link? Also, given the way of annotating these links, is it likely that due to the missing wedge, some link conformations might have been overlooked?

- Thin filament processing

It is interesting that the thin-filament map (where Tropomyosin has been masked out) has a lower resolution (8.3Å), but shows significantly higher structural detail than the 8.2 Å thin filament with Tpm masked in. Does this depend on the B-factor used for sharpening? If yes, how was the sharpening value determined (i.e. was LocSpiral also applied in thin filament processing)? Or could it be that a smaller mask used to estimate the resolution for the thin filament-only map is the reason for the increase in resolution?

Extended Data figure 1:

For consistency state the pixelsize for cryYolo filament picking also in the figure.

The 2D classification of the central slab projections was done in which software?

It might not be clear to every reader if after 2D classification, processing was again done in 3D (i.e. using Relion's subtomogram averaging implementation)? Please state explicitly.

Extended Data Figure 4a:

It is not entirely clear how the thick filament density projection is shown from a ZY and ZX view, given that the legend only mentions summing intensities along the X axis. Please clarify.

Extended Data Figure 7:

This figure is very nice. However, it is not exactly clear why in panel c in CSR7 no contact between titin and myosin is annotated, while it appears to be there in the model view on top.

Figure 2 legend

Panel d): "The complete crowns' nomenclature is indicated in". Maybe remove "in".

Florian Schur

Author Rebuttals to Initial Comments:

Point-to-point response to the reviewers' comments

We thank the reviewers for their positive feedback and insightful comments, which aided us to further improve the manuscript. We have thoroughly considered and implemented the suggested changes, which have significantly improved the quality and clarity of our manuscript. To facilitate a clear understanding of the changes made, we have highlighted them in yellow throughout the revised text.

Referee #1

Summary of the key results

The manuscript describes a FIB SEM / cryo-electron tomography study of mouse cardiac muscle in the relaxed state. The resulting structural models, particularly of the myosin thick filament, but also its interaction with surrounding thin filaments are stunning and one of the most exciting pieces of structural biology I have seen to date. The combination of sub-tomogram averaging and interpretation of raw tomograms provides an unprecedented insight into the whole structure.

Both this paper, and the co-submitted manuscript by the Craig lab, explain for the first time how different “crowns” of myosin heads have the potential to behave differently in the muscle. They also provide an elegant description of how the giant titin proteins run through the thick filament. This manuscript, by imaging the entire thick filament in its native context also provides fascinating information about the thick/thin filament interactions mediated by the MyBP-C protein and the structure of the M and P zones of the thick filament.

Originality and significance: if not novel, please include reference

The work is exceptionally novel and significant. It answers years-worth of questions regarding the organisation of myosin's in muscle and provides the basis of many studies in the future.

Conclusions: robustness, validity, reliability

The resolution of the structure is by necessity quite low in places, but the authors are extremely careful in their interpretations and are guided by both previous structures and alpha-fold. The only place where I felt their interpretation was less clear was the model they build of the titin filament. I miss, given the current resolution, how they are able to assign the Ig-like and Fn3-like domains of the titin superrepeat to the density (e.g. page 8 line 221 – needs more explanation and a figure showing density – in addition to the movie).

We have changed the methods section to better explain how we assigned the Ig-like and Fn3-like domains of titin chains. The density for titin super-repeats can be observed in Fig. 3a, 4h, and in Extended Data Fig.7a-c. In Extended Data Fig.7a-c, the densities of the two titin chains are shown after “color zone” and “split by zone” segmentation of the original reconstruction, executed in ChimeraX.

Suggested improvements: experiments, data for possible revision

Besides the above, no new experiments or analysis are required. However as described below I would recommend the manuscript and figures would benefit from some re-organisation.

Clarity and context: lucidity of abstract/summary, appropriateness of abstract, introduction and conclusions

1. Mostly the manuscript is well written and appealing to a general audience.

We thank this reviewer for this positive feedback.

2. There are a few minor grammatical errors: e.g. in the abstract “thick filaments” and “its” rather than “their”.

We carefully went through the manuscript and corrected all grammatical errors we found.

3. The one big problem with the manuscript is that the Figure and Extended Data Figure references are jumbled up and do not appear to correspond to what is being described in the text.
 - E.g. Page 7 line 174 (Fig 5a does not show what is described)
 - Page 8 & 9: Titin section: All figure references seem to be wrong/not show what is reported in the text.
 - Page 12 & 13. Same problem.

We thank this reviewer for pointing this out and have fixed all issues regarding the order of figure references.

4. The flow of the manuscript needs to be reworked slightly to follow the Figures. Currently it jumps over Fig 4a-d, straight to Fig4e.

We have reorganised the figure and panels order to better reflect the flow of the text in the revised manuscript.

5. I would bring some parts of the Titin description, currently in the Supplement, into the main text. It was very hard to follow the description without these figures.

This is a great suggestion. We have moved the panel showing the overall organization of titin domains from the Extended data in Fig. 7a (new Extended Data Fig. 6a) to Fig. 5a to make it easier for the reader to follow our descriptions.

6. Some specific figure comments: I really like the color scheme in general. It is calm and elegant.

Thank you for this nice comment.

- In Fig2f it is hard to tell the difference between Titin alpha and beta (label or change color slightly?).

We have added labels to the alpha and beta chains of in all figures to improve interpretability. In addition, we corrected the wrong colour assignment for titin in figure 2 (now Fig. 5) as was noticed by reviewer #2.

- In Fig2i the numbering 1st, 2nd etc doesn't match the text (1,2 etc). Perhaps put these cMyPB-C stripes to the left of "g" where it will be clearer what they refer to?

We have completely revised figure 2 (now Fig. 5).

- Fig 3c: Label "Blocked Head (BH)" etc to make it clearer what "BH" corresponds to.

We have changed this in the figure and text.

- Fig4: perhaps move Fig4d and 4h near to each other. Also flip 4h so it is in the same orientation? Fig4f – what is the yellow density contacting Crown1 and Crown3? Label?

We have reorganized the panels, so that panel d and h are closer to each other. In addition, we tried to incorporate the reviewer's suggestion for Fig. 4h. However, the resulting figure showed an obstructed view of the binding of cMyBP-C C10 to the Crown 2 tail array that, in our opinion, reduced the overall interpretability of the panel. Therefore, we kept it as it is. We have added the missing labels in Fig. 4f (now Fig. 3f).

- Fig6: label what the green density in Fig6c is – also refer to it in the figure legend?

We have added the description "M-band associated proteins" to both the figure and the legend.

7. Page 5, line 120: the title says the order is myosin, cMyBP-C, titin. In contrast the text goes myosin, titin, cMyBP-C.

We have corrected the order.

8. Similarly, the order of the Figures is roughly: Overview, myosin heads (3), cMyBP-C (4), myosin tails (5), M-band (6). With no Titin figure. The order of the text sections is Overview: myosin tails, myosin heads, titin, M-band, cMyBP-C. I understand it is a complicated system with lots of overlap between what needs to be talked about, but I think the manuscript will be easier to read if it follows a more fixed order and that order was followed throughout.

To reduce the discrepancy between the figures and the order of the paragraphs, we have moved the titin figure from Fig. 2 to Fig. 5.

9. Two very minor points:
 - a. Could the authors explain explicitly that the "interacting heads motif" is actually a structural conformation of the myosin molecule (rather than a peptide motif).

Good point – we have added this to the revised manuscript.

- b. I understand the differences in nomenclature between this paper and the Craig paper (Titin L/R vis alpha/beta), Crown 1-3 vis CrH,CrT,CrD etc. It is ok to leave it, but I wonder if there is any chance the authors could confer and come up with a unified nomenclature?

We agree with the reviewer and also believe a common nomenclature would be beneficial to the readership. Our nomenclature for titin is based on the observation that titin alpha is the full-length chain, while titin beta is the chain missing the C-terminus domains. For the crowns' nomenclature, we follow the numbering system of a 3-start helix that starts with the first crown from the M-band. This nomenclature is independent of the orientation of the reconstructions and is consistent with the one used in previous structural investigations of the vertebrate thick filament (Al-Khayat et al., 2012).

Referee #2

This paper presents the discovery of the three-dimensional structure of a major part of the mouse cardiac thick filament in the relaxed state. The structures of the myosin, titin and cMyBP-C molecules can all be seen. The 3D arrangement of all three molecules that is revealed include unpredicted features that mean that this work is not merely an extension of previous studies of invertebrate thick filaments but represent a completely new and more complex arrangement than hitherto suspected. All previous models of vertebrate thick filament structure are proven wrong by these new results. A structure for the blocked state thin filament in situ is also a new result of this study. As the work is done on mammalian cardiac myofibrils, the results also have implications for the understanding and treatment of many cardiac diseases, so the results will be of interest to an unusually wide audience.

The results are obtained by an established and reliable workflow, the text is lucid and the Figures effectively communicate the complex results.

The benefits of using cryo-ET on intact myofibrils compared to working with isolated thick filaments are evident from the ability to characterise the interactions of cMyBP-C with both thick and thin filaments as well as by the discovery of perturbations in the structure along the length of the filament that would easily have been missed by averaging the repeating 43nm units. However, this does limit the resolution obtained to about 2 nm, which does not reveal secondary structure within the proteins, only their overall shape, and the coiled coil tail of myosin is a smooth cylinder so that the local azimuth of its coiled coil structure cannot be seen. To interpret their results in terms of an atomic model of the structure, the authors make sensible decisions in modelling in known structures and AlphaFold predictions, but it must be recognised that, for instance the orientation of titin and cMyBP-C domains within the blobs of density are best guesses, and are not determined by the EM data itself. For the benefit of the general readership of Nature, the authors should include some such statement at the start of the Results section.

AlphaFold2 models showed that various parts of the prediction had relatively rigid conformations that greatly helped the fitting of the domains. We now describe in the main text as well as in the methods, that and how we used AlphaFold2 to create the model.

Video 1 shows features of the myofibril specimen that raise some concerns. It shows that the thick filaments are not in transverse register: the bare zones are axially staggered by up to ~300nm in the sample shown, whereas they should be aligned. This indicates that the filaments actively sheared past one another during specimen preparation, probably at the step at which ATP was added to the rigor myofibril preparation, as this induces a brief contraction. A consequence is that the M-line bridges will have been broken and that suggests that the M-line structure reported is not the native one. Likewise, there are thin filaments throughout the bare zone, indicating that the sarcomere length is short, and there is little indication that the thin filaments are located at the trigonal points of the thick filament array as they should be. In consequence thin filaments of the opposite polarity may be present alongside the thick filaments and some of the cMyBP-C bridges described may involve such filaments and are thus likely to be non-physiological. To address these concerns, the authors should report the sarcomere length of the specimens they have imaged, and make counts of the thin to thick filament ratio in their tomograms – is it 2:1 as in vivo, or 4:1 as in over-shortened myofibrils? These concerns do not however undermine the great advances that are presented on thick filament structure.

The plane of the sarcomeres and the angle of the milling beam are rarely parallel to each other, resulting in a reconstructed tomogram with filaments tilted with respect to the XY plane. This could lead to the optical illusion of a strong staggering.

Nevertheless, we understand the reviewer's concerns about the staggering and the presence of thin filaments in the bare zone and thank him very much for his excellent suggestions. Indeed, while most of the M-band, P-zone, and C-zones align nicely, we also observed slight longitudinal shifts in several regions. In addition, we did not observe a clear hexagonal arrangement of filaments in all regions of the tomograms and found that some thin filaments enter the M-band and some actin filaments are very long and stretch over the M-band into the contralateral half-sarcomere.

The loss of a clear hexagonal arrangement and some longitudinal shifts is probably due to the fact that we used demembrated myofibrils, which lack the intracellular and extracellular osmotic pressure and cytosolic proteins that laterally stabilizes myofibrils also in the relaxed state.

When working with cardiac tissue, we usually observe that filaments enter the bare zone and M-band and even cross over to the contralateral half-sarcomere. This is independent of the sarcomere length. This phenomenon is physiological and was already described by Robinson and Winegrad (*J Physiol* 286: 607-619. 1979). It was later shown to be due to different lengths of thin filaments in cardiac tissues that do not express nebulin (Littlefield, R., et al. (2001). *Nat Cell Biol* 3(6): 544-551; Kolb, J., et al. (2016). *J Mol Cell Cardiol* 97: 286-294.)

As proposed by the reviewer, we measured the sarcomere length and the thin-to-thick filament ratio to prove that the presence of thin filaments in the M-band is not caused by

hypercontraction. The average sarcomere length of the dataset was $2.26 \mu\text{m}$ (S.D.= $0.11 \mu\text{m}$ - $N=45$), indicating that the sarcomeres are indeed not hypercontracted. This is also supported by the 1.95:1 (102 thin filaments, 53 thick filaments) thin-to-thick-filament ratio in the two tomograms we segmented. In agreement, with our observation that thin filaments cross the M-band, the thin-to-thick-filament ratio is higher in the P-zone (2.3:1). Therefore, we are confident that although some M-bridges may be broken in our tomograms, the sarcomeres are not hypercontracted and associated artifacts can be excluded. In addition, when characterizing the cMyBP-C links, we were careful to include only the regions that contained thin and thick filaments of the same half-sarcomere. The revised manuscript contains an appropriate statement.

Minor points:

There is some confusion in labelling of the two titin chains. In the text (line 230) it is stated that titin alpha runs across the bare zone, but in Fig. 2(f) the label Titin Alpha is dark pink, corresponding to the colour of the titin chain that ends near crown A1. Also in panel (g), the truncated chain appears to be the paler (left-hand) one, i.e. the opposite colouring of panel (f). The authors need to correct this confusion, but also to check every Figure panel to be sure that the colouring of both the protein and the labelling are consistent throughout the paper. Given this mix-up of colouring, they should also check every text statement to make sure that statements about alpha and beta titins do indeed refer to the correct molecule, rather than a statement linked to an incorrectly-coloured molecule. It would also help the reader if the difference in shade of pink of the two titins and all the titin labels was made more pronounced in all the Figures.

We have corrected the errors in the color assignment and added labels and models to improve interpretability in Figure 2 (now Fig. 5).

Related to the last point, the use of depth-dependent colouring makes a nonsense of the distinction between the two titin chains, because the dark pink titin chain becomes pale pink when it is further away. This renders ED Fig. 5b and, especially, ED Fig 9 almost impossible to decipher. Please turn off depth cueing and replace any panels in which it has been used.

We have removed depth cue in all the figures where it wasn't strictly required. We also reduced the silhouette's thickness in the models to better preserve the contrast between the different shades and tints of pink. In ED Fig.4 c-j we use now transparency to allow for the visualization of the underlying Coulomb surface colours.

The labelling in the otherwise excellent Fig. 2 should be revised to reduce confusion. In panel (d) the space between successive crown numbers is variable, which I initially thought reflected an actual variation in crown spacing in panel (c), but the correlation is not good. It would be helpful if the numbers were indeed carefully placed so that they did emphasize variable crown spacing, for instance by placing them such that the lower edge of each number was at exactly the same level as the lever-lever junction of each IHM crown. The titin labels (panel (h)) should be changed from C2-C11 to CSR2-CSR11, like in ED Fig7, to reduce confusion with C-stripe

numbers. It would also help to change them from prominent black to pink, which will help the reader to notice the relatively visually weak ochre-coloured C-stripe labels to the right (panel (i)). Finally, panels (g-i) are at a slightly larger scale than the rest of the Figure, such that they are in register with panel (f) at the top of the Figure, but are out of sync at the bottom. Please adjust.

We have realigned and adjusted the spacing of the numbers of the crowns in Fig. 2d (now Fig. 5) so that they are aligned to the same points used to calculate the Euler angles of the heads and cylindrical coordinates (the centroid of the triangle formed by BH, FH, and S1/S2 junction). We have also changed the labelling for titin in panel j from C2-C11 to CSR2-CSR11 to reduce confusion with C-stripe numbers as suggested. To further reduce the risk of confusion with the labelling of cMyBP-C stripes, we have changed the order of the panels as suggested by reviewer #1. The scaling between different panels has now been corrected, too.

Video 2 fails to colour the two titin chains differently in the EM density surface, to match Fig. 2. In correcting this, it won't matter if one of them is then the same colour as the tropomyosin strands, as there's no chance of confusing the two structures.

We have corrected Supplementary Video 2 accordingly.

Line 230: Apart from a few blobs which might be titin or some other myofibrillar protein, I don't see any evidence of titin in the bare zone in Fig 2 (a, b), let alone that it crosses all the way to the opposite P-zone, so what is the basis for the assertion that the alpha chains do this? Is this a conclusion from some other (unreferenced) study by antibody labelling? On a more general point, is the truncation of beta titin that they observe a consequence of the shearing between thick filaments that has disrupted M-line structure? It is possible that beta titin was attached to M-bridges and therefore was snapped apart by shear, or was more susceptible to postmortem proteolysis in the bare zone than alpha titin molecules, with the result that the clump of titin domains near A1 is an artefactual recoil rather than a native structure.

The assertion of titin spanning over the M-band does indeed come from previous antibody labelling studies (Obermann et al., 1996). We now cite this study in the revised manuscript. Additionally, we recently obtained unpublished structural data on intact and aligned M-bands that confirms the presence of the titin bow tie shaped structure in the P-zone. This rules out that the titin beta conformation is a preparation artefact and supports its physiological significance.

Location of the titin kinase (TK) domains: By using antibody labelling the site of TK is measured to be 79 nm from the centre of the bare zone. The authors state that this is the position of the TK of the alpha chain, and that the TK of the beta chain is unlabelled. But the putative densities in their EM density map are only about 43nm apart. It is doubtful that their STED imaging is capable of resolving two positions that are so close, since there is only a narrow separation between the two stripes separated by 156 nm across the bare zone. Therefore, how can they rule out that the antibody has labelled both TK domains and the separation measured by STED is the midway point between the alpha and beta copies of TK across the bare zone? If the truncation of beta titin is an artefact (see comment above), this STED result might reflect where the beta titin TK was before M-line disruption.

We agree with the reviewer that STED microscopy, especially considering the lower resolution in the Z direction and the thickness of myofibrils of $\sim 1 \mu\text{m}$, is unlikely to resolve the distance of 43 nm in a conventionally immunostained sample. We do not infer from our data, however, that we see *only* the labelled alpha chain (although epitope accessibility of the two TK domains might be different). The labelling is most consistent with the position of the alpha chain TK, but at this resolution it may actually be a composite signal from both TK domains albeit at different intensities. This caveat does not invalidate the interpretation that the TK domain(s) are located at P1 and A1 and not at the edge of the M band, where they were placed in the previous immuno-EM by Obermann et al.

We have clarified this limitation in the description of the results as follows:

"This is most consistent with titin kinase density at the P1 crown, but does not preclude localization of TK at both A1 and P1 crowns (43 nm apart) given the expected resolution limits of STED microscopy."

Lines around 360: Because the thin filament repeats at $\sim 38\text{nm}$ intervals and the cMyBP-C at 43nm intervals, is this flexibility due to the cMyBP-C always seeking out the same target on blocked state thin filaments? - i.e. do the attachment sites of cMyBP-C on thin filaments show a 38nm periodicity?

The reviewer correctly points out that we forgot to mention this aspect. Although the resolution does not allow a definite statement, the cMyBP-C seems to follow the symmetry of the thin filament similar to the myosin heads in the rigor state. Thus, the flexibility of the links makes likely allowance for the mismatch of the 37-nm actin repeat in thin filaments and the 43-nm cMyBP-C repeat in thick filaments. This is now included in the revised manuscript.

Line 52: Myosin molecule mass should include the masses of its four light chains, not just its two heavy chains, and therefore be around 530 kDa not 446 kDa.

This is correct and we have changed it accordingly in the revised manuscript.

Line 118: the claim of describing '...the thick filament across the entire sarcomere...' is unjustified as the D-zone is sadly not included in the current study.

We have corrected this statement and hope to be able to present the structure of the D-zone in a follow-up study.

Fig. 3 legend needs to state the direction of view of the cross sections – M-wards or Z-wards?

This is an excellent suggestion. We have added indications for directionality in Fig. 3 and several other figures.

Line 225: Figure referenced should be Fig 2b not 5b.

We have corrected this typo.

The legend to ED Fig. 9 (d-f) is too brief – it doesn't say what each of the three panels depicts.

We have extended the legend accordingly (now Extended Data Fig. 8).

Lines 266 and 269: Here there is reference to a bow-tie feature, supposedly depicted in ED Fig 9d. I think this should be a reference to panel (c) of the Figure in which there is a label 'Titin beta tie', but even there what I see to the left of the label is a U-shaped sequence of domains, not a bow-tie shape. Maybe 'cluster' would be simpler.

The figure has now been revised and includes two schematics to help the reader understand the topology of titin, particularly in the bow-tie region of titin beta.

Lines 329 and 856: 'interphase' should be 'interface'.

We have corrected this typo.

Referee #3

Tamborrini and colleagues use advanced cryo-electron tomography methods to report the organization of the muscle thick filament in its relaxed state in situ. This allows them to provide a holistic description of the components forming the thick filament in their native context, providing yet unknown insights into their interaction during muscle contraction. Together, with other recent studies from the same lab (Wang et al., 2021 & Wang et al., 2022) the authors continue to fill in a complete high-resolution structural picture of the sarcomere architecture. The wealth of biological observations obtained make this a very complex and fascinating study. I applaud the authors in preparing figures that do a good job in guiding the reader through the different aspects of the complex thick filament structure.

The methodological implementation of their established cryo-focused ion beam milling and cryo-ET pipeline is again state-of-the-art and meticulously performed. My review will focus on technological aspects which deserve attention for minor clarifications.

Overall, this is an outstanding study, considering both the biological findings and methodological implementation.

Thick filament processing:

- In Extended Data figure 1 the mentioned differences in crowns of classes A, B and C are difficult to see (projected, retracted IHMs in class A and C, and the fuzzy appearance for crown 2 IHM in class B). Please annotate these differences.

We have added a window showing a magnified view to highlight these differences in crown 2.

- It is also not entirely clear how these different classes are reflected in the different obtained structures. Specifically, class A apparently showed a unique distribution in the sarcomere. I assume this means class B and C did not? Does this reflect something of biological relevance or has this been caused by any limitation in the classification process (e.g. lower SNR of such particles)?

The reviewer correctly concluded that in the first level of classification, only Class A, which contains a projected IHM at crowns 2, has a unique distribution in the sarcomere. When we projected the 3 classes back into the tomogram, the coordinates of classes B and C showed no particular distribution. However, all class B particles had a similar distance from the M line, suggesting that they were representatives of a particular section of the thick filament. Importantly, before finding this processing pipeline, all the 3D classifications we run without helical symmetry failed to distinguish any specific thick filament segment from the averaged reconstructions.

We now know that using helical symmetry in 3D classification works because it exploits the similarity of two consecutive crowns 2 in a given segment.

In our first 3D classification, we used helical reconstruction to allow initial alignment of thick filaments and to determine their polarity. By imposing helical symmetry, we averaged out the heterogeneity of the different crowns, leading to classes B and C being a conglomerate of all different segments and showing poorly resolved IHMs. However, crowns 2 of class A showed sufficient similarity to classify them into a separate class with "projected" IHMs. It later became clear that the crowns 2 that contributed to the helical average of class A were crowns A5 and A8. As it turned out, while all crowns 2 have a beta angle of $\sim +30^\circ$, crowns A5 and A8 have a beta angle of $\sim -25^\circ$. We believe this explains why the "projected" class stood out during the refinement with imposed helical symmetry.

We describe this now better in the Methods of the revised manuscript.

- In particular, given another recent study on the cardiac myosin filament (Dutta et al.), where one of the crowns appears to be disordered, one could speculate about whether this structure corresponds to one of the observed classes in the complete in situ thick filament.

Prior to identifying all different CSR segments of the thick filament, our initial maps contained unsorted segments and showed a similarly unresolved crown 2 density (see Class B). After further classification however, this was not the case, indicating that the disordered appearance of the crowns in the Dutta et al. study is likely due to averaging different conformations of crown 2.

It is not entirely clear what the authors refer to with "Through an iterative process of 3D refinement, multi-reference 3D, classification, particles back-projection, and axial shift calculations" (Methods, page 22/line 635). Can the authors explain better what the axial shift rotation calculations are and why they were necessary?

The calculations for the axial shift of the coordinates, allows to shift the coordinates of a filament within the tomogram along its z-axis. Given the coordinate and the orientation of a particle, we calculated the position of the neighbour segments on the same filament in order to extract sub-volumes belonging to the same CSR of the thick filament. We have rewritten this part of the Methods section and added more details.

Also, I assume the processing has been done via gold-standard dataset splitting. Could the authors elaborate at which step in the processing the datasets have been separated. This is

not an crucial comment, but still would be interesting to know, given the shape of some FSC curves, that show a jump close to the 0.143 criterion (e.g ED Figure 2c,d,i).

After classification and re-extraction, all 3D refinements were performed in RELION 3.1, assuming a featureless cylinder and splitting the particles into two independent groups from the first iteration, corresponding to the gold standard dataset splitting procedure. We believe that the shape of the FSC curves in Extended Data Fig. 2 c,d,i reflects the intrinsic properties of these three maps.

The way how the authors have segmented the cMyBP-C links is very creative and clever. However, it would be interesting to know what kind of validation was used to verify that a link between thick and thin filaments has been identified accurately, i.e. was after segmentation every remaining non-assigned density in the tomograms between thick and thin filament automatically assigned to be a link?

For the validation of the links, our original plan was exactly the one the reviewer is suggesting: after resolving all possible structures in the tomographic volumes, we would use our pseudo-segmentation procedure to subtract the signal from the tomogram – the remaining non-assigned density would be annotated as links. Unfortunately, when we implemented this approach, we had to use binned tomograms to not run in hardware limitation. The use of binned reconstruction and binned coordinates resulted in minor mismatches that generated a constellation of “non-assigned densities”. Considering that a single Ig-like domain would only be about 4x2x2 voxels, it was difficult to find parameters that could differentiate the mismatched voxels from the actual signals of the links. We eventually had to forfeit the automatization of this procedure and resort to user-based segmentation. With this approach, only densities that belonged to clearly identifiable globular domains were segmented. As it can be observed in Fig. 3e-f, after segmenting the links, virtually all density is assigned to sarcomere components.

To further cement the validity on this segmentation, we compared the distances from the M-line of the 9 putative C7 cMyBP-C domains with previously published immune electron microscopy studies (Tonino et al., 2019) and found perfect consistency with their measurements. Finally, the presence of flexible linkers, can also be observed in the cross-section of our reconstruction as low threshold densities in Extended Data Fig. 9a.

We have added these explanations to the revised manuscript.

Also, given the way of annotating these links, is it likely that due to the missing wedge, some link conformations might have been overlooked?

We agree with the reviewer: due to the missing wedge, some links have likely been overlooked, specifically those more aligned to the Z axis in the tomography volume. However, considering that the thick filament has a three-fold rotational symmetry, we can assume that the segmented group is representative of the entire population.

Thin filament processing

It is interesting that the thin-filament map (where Tropomyosin has been masked out) has a lower resolution (8.3Å), but shows significantly higher structural detail than the 8.2 Å thin

filament with Tpm masked in. Does this depend on the B-factor used for sharpening? If yes, how was the sharpening value determined (i.e. was LocSpiral also applied in thin filament processing)? Or could it be that a smaller mask used to estimate the resolution for the thin filament-only map is the reason for the increase in resolution?

For the thin-filament map and the F-actin map we applied a B-factor of -200 and -100, respectively. These values were empirically determined based on visual inspections of the maps. We mention them now in the revised manuscript. The better resolution but less detail in the thin filament reconstruction may be due to the use of helical symmetry for this reconstruction but not for the F-actin reconstruction, which often leads to a slight overestimation of the resolution. On the other hand, as the reviewer suggested, this can also likely be a mask-dependent phenomenon.

Extended Data figure 1:

For consistency state the pixelsize for cryYolo filament picking also in the figure.

We state this now in the revised manuscript.

The 2D classification of the central slab projections was done in which software?

It was done using ISAC. We now explicitly state this in the Methods section.

It might not be clear to every reader if after 2D classification, processing was again done in 3D (i.e. using Relion's subtomogram averaging implementation)? Please state explicitly.

We now explicitly state this in the Methods section.

Extended Data Figure 4a:

It is not entirely clear how the thick filament density projection is shown from a ZY and ZX view, given that the legend only mentions summing intensities along the X axis. Please clarify.

Thanks for pointing this out. We forgot "or the y axis". This has been added in the revised manuscript.

Extended Data Figure 7:

This figure is very nice. However, it is not exactly clear why in panel c in CSR7 no contact between titin and myosin is annotated, while it appears to be there in the model view on top.

Thank you for this nice comment. Although it looks from this perspective as if there were an interaction between CSR7 and crown 2 myosin tail, this is not the case and a perspective deception. We tried to use depth cue to avoid this effect. However, this made it difficult to interpret the figures (see also reviewer #2). Therefore, except for changing the color of titin, we kept the figure as it is.

Figure 2 legend

Panel d): "The complete crowns' nomenclature is indicated in". Maybe remove "in".

This has been done.

Reviewer Reports on the First Revision:

Referees' comments:

Referee #1:

The authors have addressed all my concerns as well as those of other reviewers. I strongly recommend its publication.

Referee #2:

From the reader's point of view, it is regrettable that the two groups (Tamborrini and Dutta) have not reached a mutually agreed terminology for identical components of the thick filament structure and have conflicting colour schemes for the crowns, and this is despite both groups agreeing with Referee #1 that harmonisation is important. This will be a source of great confusion for everyone and thus cause error in future, and is thus important to avoid. Given the existence of online conferencing, and for the sake of this research field, can the two groups work together to make this happen?

The authors have addressed the referees' comments adequately. They have also taken the opportunity to revise many Figures to improve readers' comprehension, and to correct errors.

Some minor corrections I spotted and checking needed:

Line 46: the convention for naming in myosins is for the 'head' to consist of the motor domain plus the light-chain binding domain. Thus myosin-2 has two heads that attach directly to the tail. The IHM is thus formed by the two heads plus the proximal tail. Conflating 'head' with 'motor domain', and therefore using the additional term 'neck' to refer to the light chain binding (lever) part of the conventional head adds a layer of ambiguity, especially as the Dutta paper uses 'head' in the conventional sense, and I think you have used 'head' to mean motor domain plus lever throughout the manuscript. Therefore please change the definitions here to the conventional ones, getting rid of 'neck', and check the manuscript for any instances where 'head' needs to be changed to 'motor domain' to communicate the intended meaning.

Note that the mean sarcomere length in the Responses Letter is given as 2.26 and in the revised article line 621 as 2.326. Please ensure the correct value is used in the final article.

Fig 2d-i: add statement in legend whether these are viewing M-wards or Z-wards (I think the latter). And check that the direction of view is clearly stated for cross-sections in all other Figures too. It would help in relating the appearance of these cross-sections (d-i) to the side views (a, b) if each of them was rotated by 180 degrees, so that they would be what was seen if the filament as oriented in a,b was rotated 90 degrees around a horizontal axis so one was looking Z-wards.

Fig 4b legend: Does 'top view' mean looking down from the top (ie looking Z-wards) or up towards the top (ie looking M-wards)? It would be unambiguous to use the 'looking M/Z-wards' form of words.

Fig 6c is not the same side view as Fig 6a, since the members of the P1 crown appear axially staggered, suggesting that the filament long axis does not lie in the plane of the Figure. Please correct this.

ED Fig 5 b: the arrow relating left and right views is pointing the wrong way (it is a left-handed rotation).

ED Fig. 6 and Legend: To cut down on needless abbreviations, please replace 'S1/S2 (site)' with 'head-tail junction'. Also in ED Fig 7 legend.

ED Fig 7a: I could not work out from the Methods and legend's texts the relationship between the IHM and the imaginary reference equilateral triangle that is mentioned. I did not understand the Methods statement '...has its top side parallel to the plane of the cylinder...' Which side of the (non-equilateral) triangle of the IHM is being compared to which side of the reference triangle? A small explanatory diagram would be very helpful that showed the cylindrical filament, reference triangle and an example IHM. It would also be useful if the diagram also answered the following series of questions: Is $\alpha=0$ parallel to the filament long axis, is $\beta=0$ a radial line and is $\gamma=0$ a tangent to the cylindrical surface? I spotted that in the relevant part of Methods, references are made to ED Fig 8 instead of ED Fig 7, which needs correcting, and a thorough check made that the correct revised Figures are referred to throughout the MS. At line 725 'angels' should be replaced by 'angles'.

In ED Fig 7d what is the meaning of the variable leftward leaning of the arrows? I'm guessing it is something to do with the 'rise' parameter, but as the x-axis has no scale, it is not apparent what is being plotted here. It would be helpful for this Figure's legend to be more verbose, such that the reader can readily understand all parts of the Figure.

ED Fig 9 Are the panels in (a) looking Z-wards? Please add labels on the left-most panel to indicate one of the cMyBP-C and its NA-TF and PA-TF within the surrounding annulus of density. Panel a would probably benefit from inverting the contrast, so the (rather weak) grey density of the cMyBP-C stands out against a white background rather than being lost into the blackness. In panel (c), I don't understand in what sense the intensity profile is specifically that of a top view – isn't it just a radial intensity profile?

ED Fig 10 panel c has been changed to be rotated by 180 degrees compared to panel b, but the linking arrow still states 120 not 180.

Line 572: 'interphase' should be 'interface'.

Referee #3:

The authors have addressed the issues brought up in my review of the first submission. I have no further concerns or comments and recommend the manuscript for publication.

Florian Schur

Author Rebuttals to First Revision:

Point-to-point response to the reviewers' comments – round 2.

We are extremely grateful to all three reviewers for their positive feedback. Their comments aided us to improve the quality and readability of our manuscript. We are happy that all three reviewers now find the manuscript acceptable for publication. Below is a response to final suggestions from reviewer #2.

Referees' comments:

Referee #1:

The authors have addressed all my concerns as well as those of other reviewers. I strongly recommend its publication.

Referee #2:

From the reader's point of view, it is regrettable that the two groups (Tamborrini and Dutta) have not reached a mutually agreed terminology for identical components of the thick filament structure and have conflicting colour schemes for the crowns, and this is despite both groups agreeing with Referee #1 that harmonisation is important. This will be a source of great confusion for everyone and thus cause error in future, and is thus important to avoid. Given the existence of online conferencing, and for the sake of this research field, can the two groups work together to make this happen?

We have coordinated with Dr Craig to provide a clear guide in each paper showing the mutual correlation between the two groups' color schemes and terminology. This is shown in a table in each paper. This will avoid any possible confusion.

The authors have addressed the referees' comments adequately. They have also taken the opportunity to revise many Figures to improve readers' comprehension, and to correct errors. Some minor corrections I spotted and checking needed:

Line 46: the convention for naming in myosins is for the 'head' to consist of the motor domain plus the light-chain binding domain. Thus myosin-2 has two heads that attach directly to the tail. The IHM is thus formed by the two heads plus the proximal tail. Conflating 'head' with 'motor domain', and therefore using the additional term 'neck' to refer to the light chain binding (lever) part of the conventional head adds a layer of ambiguity, especially as the Dutta paper uses 'head' in the conventional sense, and I think you have used 'head' to mean motor domain plus lever throughout the manuscript. Therefore please change the definitions here to the conventional ones, getting rid of 'neck', and check the manuscript for any instances where 'head' needs to be changed to 'motor domain' to communicate the intended meaning.

We changed the nomenclature accordingly throughout the manuscript and removed the reference to the 'neck'.

Note that the mean sarcomere length in the Responses Letter is given as 2.26 and in the revised article line 621 as 2.326. Please ensure the correct value is used in the final article.

2.326 μm is the correct average sarcomere length and we made sure that it is written as such in the revised article.

Fig 2d-i: add statement in legend whether these are viewing M-wards or Z-wards (I think the latter). And check that the direction of view is clearly stated for cross-sections in all other Figures too. It would help in relating the appearance of these cross-sections (d-i) to the side views (a, b) if each of them was rotated by 180 degrees, so that they would be what was seen if the filament as oriented in a,b was rotated 90 degrees around a horizontal axis so one was looking Z-wards.

We have revised the figure and legend accordingly.

Fig 4b legend: Does 'top view' mean looking down from the top (ie looking Z-wards) or up towards the top (ie looking M-wards)? It would be unambiguous to use the 'looking M/Z-wards' form of words.

We have revised the legend accordingly.

Fig 6c is not the same side view as Fig 6a, since the members of the P1 crown appear axially staggered, suggesting that the filament long axis does not lie in the plane of the Figure. Please correct this.

We have corrected this.

ED Fig 5 b: the arrow relating left and right views is pointing the wrong way (it is a left-handed rotation).

This has been corrected.

ED Fig. 6 and Legend: To cut down on needless abbreviations, please replace 'S1/S2 (site)' with 'head-tail junction'. Also in ED Fig 7 legend.

We have corrected this.

ED Fig 7a: I could not work out from the Methods and legend's texts the relationship between the IHM and the imaginary reference equilateral triangle that is mentioned. I did not understand the Methods statement '...has its top side parallel to the plane of the cylinder...' Which side of the (non-equilateral) triangle of the IHM is being compared to which side of the reference triangle? A small explanatory diagram would be very helpful that showed the cylindrical filament, reference triangle and an example IHM. It would also be useful if the diagram also answered the following series of questions: Is $\alpha=0$ parallel to the filament long axis, is $\beta=0$ a radial line and is $\gamma=0$ a tangent to the cylindrical surface? I spotted that in the relevant part of Methods, references are made to ED Fig 8 instead of ED Fig 7, which needs correcting, and a thorough check made that the correct revised Figures are referred to throughout the MS. At line 725 'angels' should be replaced by 'angles'.

We agree with the reviewer that ED Fig 7 is quite convoluted and could be improved. We have modified the method section to better describe the process of calculating Euler angles and cylindrical coordinates for each crown. In addition, as suggested, ED Fig.7 now contains a diagram representing the Euler angles

calculated for crown2 (A5) as a representative case that can help the reader better understand the rest of the figure. The legend has also been expanded to clarify the concepts.

In ED Fig 7d what is the meaning of the variable leftward leaning of the arrows? I'm guessing it is something to do with the 'rise' parameter, but as the x-axis has no scale, it is not apparent what is being plotted here. It would be helpful for this Figure's legend to be more verbose, such that the reader can readily understand all parts of the Figure.

ED Fig 7d contains 2 stacked charts, sharing the same x-axis. To visually clarify this, we extended the y-axis so that it would encompass and connect both charts to the same x-axis. Additionally, the legend has also been amended to help the interpretability of the content.

ED Fig 9 Are the panels in (a) looking Z-wards? Please add labels on the left-most panel to indicate one of the cMyBP-C and its NA-TF and PA-TF within the surrounding annulus of density. Panel a would probably benefit from inverting the contrast, so the (rather weak) grey density of the cMyBP-C stands out against a white background rather than being lost into the blackness. In panel (c), I don't understand in what sense the intensity profile is specifically that of a top view – isn't it just a radial intensity profile?

We inverted the contrast of ED Fig9a, added a descriptive title to indicate the view directionality, and provided the labelling for cMyBP-C, NA-TF and PA-TF. We also changed the title of ED Fig 9c to "Radial intensity profile".

ED Fig 10 panel c has been changed to be rotated by 180 degrees compared to panel b, but the linking arrow still states 120 not 180.

This has been corrected.

Line 572: 'interphase' should be 'interface'.

This has been corrected.

Referee #3:

The authors have addressed the issues brought up in my review of the first submission. I have no further concerns or comments and recommend the manuscript for publication.

Florian Schur